# Force transmission by retrograde actin flow-induced dynamic molecular stretching of Talin

Sawako Yamashiro [1,2] ✉, David M. Rutkowski [3], Kelli Ann Lynch[3,4], Ying Liu[1], Dimitrios Vavylonis [3] & Naoki Watanabe [1,2]

Force transmission at integrin-based adhesions is important for cell migration and mechanosensing. Talin is an essential focal adhesion (FA) protein that links F-actin to integrins. F-actin constantly moves on FAs, yet how Talin simultaneously maintains the connection to F-actin and transmits forces to integrins remains unclear. Here we show a critical role of dynamic Talin unfolding in force transmission. Using single-molecule speckle microscopy, we found that the majority of Talin are bound only to either F-actin or the substrate, whereas 4.1% of Talin is linked to both structures via elastic transient clutch. By reconstituting Talin knockdown cells with Talin chimeric mutants, in which the Talin rod subdomains are replaced with the stretchable β-spectrin repeats, we show that the stretchable property is critical for force transmission. Simulations suggest that unfolding of the Talin rod subdomains increases in the linkage duration and work at FAs. This study elucidates a force transmission mechanism, in which stochastic molecular stretching bridges two cellular structures moving at different speeds.

Focal adhesions (FAs) are cell-matrix adhesion structures, in which integrin connects to actin filaments (F-actin) via actin-binding adaptor proteins. Talin is a key adaptor protein regulating FA assembly[1,2]. Force induces unfolding of the Talin C-terminal rod domain, which acts as a mechanosensor[3–6]. Talin simultaneously binds integrins and F-actin. The N-terminal head domain binds integrin and the C-terminal rod domain contains two actin-binding sites, ABS2 and ABS3[7]. The Talin rod domain contains 13 α-helical bundled subdomains (R1–R13)[3,7]. Forces in the piconewton range unfold the Talin rod subdomains, thereby exposing cryptic binding sites for vinculin[6,8,9]. Talin has been proposed to function as a molecular clutch that links the actin network to integrins[2,10,11].

A stepwise mechanism is widely accepted for Talin's mode of action during FA maturation[4,8,12]. First, Talin binds integrin and F-actin. The Talin rod subdomains are stretched by forces of the retrograde actin flow and actomyosin contraction. Vinculin binds the unfolded Talin rod subdomains and F-actin, which strengthens the linkage between F-actin and integrins. After the seminal super-resolution microscopy study by ref. 13, the currently prevailing view is that the majority of Talin exists at FAs in a fully stretched state between F-actin and integrins[2,11]. On the other hand, Talin is reported to move at ≈70% of the speed of the retrograde actin flow on average in FAs of PtK1 cells[14]. In this previous study[14], the Talin velocity was measured by quantitative fluorescent speckle microscopy, in which the movement of fluorescent speckles containing multiple-fluorophore probes is measured[15]. Therefore, the velocity of individual Talin molecules has remained unclear. In contrast, the actin flow velocity has been measured by using Single-Molecule Speckle (SiMS) microscopy[16], which reports the velocity of individual molecules with nanometer-scale accuracy. Our previous study revealed that the F-actin network continuously moves as a single unit over FAs at ~20 nm/s in *Xenopus* XTC cells[16]. Thus, it is not immediately clear how Talin can simultaneously

[1]Laboratory of Single-Molecule Cell Biology, Kyoto University Graduate School of Biostudies, Kyoto, Japan. [2]Department of Pharmacology, Kyoto University Graduate School of Medicine, Kyoto, Japan. [3]Department of Physics, Lehigh University, Bethlehem, PA, USA. [4]University of South Florida, Tampa, FL, USA. ✉e-mail: yamashiro.sawako.5c@kyoto-u.ac.jp

maintain the connection to the flowing actin network and transmit forces to integrins[17,18].

Currently, only vinculin has been identified as binding to the Talin rod domain in its unfolded state[19]. On the other hand, vinculin knockout mouse embryonic fibroblasts form mature FAs, indicating that vinculin is not essential for FA growth[20,21]. Unfolding of the Talin rod domain may have additional vinculin-independent roles for FA growth and force transmission.

Using SiMS[16], our present study demonstrates that Talin exhibits a dynamic transition between flowing and stationary motions. Furthermore, Talin shows jumping and back-and-forth motions (up to 140 nm) along the retrograde actin flow, suggesting that unfolding of Talin is associated with a transient linkage between F-actin and integrin. Our experimental data and simulations demonstrate that molecular elasticity and stochastic coupling are necessary and sufficient to transmit the F-actin flow force to integrins. This study thus elucidates a mode of force transmission between two cellular structures moving at different speeds.

## Results

### Talin exhibits flowing and stationary motions

We first investigated the molecular behavior of *Xenopus laevis* Talin1 (xTalin1) in XTC cells spreading on glass coverslips doubly coated with poly-L-lysine and laminin (Fig. 1a). Under this condition, cells form thin elongated FAs in lamellipodia (Fig. 1a), which do not locally interfere with the retrograde actin flow[16,22]. xTalin1 SiMS exhibited either flowing or stationary motion in lamellipodia (Fig. 1b, Supplementary Movie 1). EGFP-xTalin1 SiMS occasionally exhibited a switching motion between the two states (Fig. 1b). We classified xTalin1 SiMS behavior into four groups: flowing (35.9%); stationary (50.0%); switching (10.3%); and unclassified (3.86%) within a 120-second time window (Fig. 1c, d, Table S1). Although xTalin1 SiMS were highly accumulated in FAs marked by mPlum-paxillin (Fig. 1a, d), all flowing, stationary, and switching xTalin1 SiMS were observed throughout lamellipodia (Fig. 1d, Supplementary Movie 1). The C-terminal EGFP fusion of xTalin1, xTalin1-EGFP, displayed similar SiMS behaviors to those of EGFP-xTalin (Table S2). Vinculin SiMS exhibited similar behavior to xTalin1 SiMS; i.e., vinculin SiMS showed either flowing, stationary, or switching motions in lamellipodia of XTC cells cultured under the same conditions. The detailed SiMS analysis of vinculin will be reported elsewhere.

To compare the velocity of flowing xTalin1 SiMS to the retrograde actin flow at single-molecule resolution, we performed dual-wavelength SiMS microscopy for xTalin1-EGFP and CF680R labeled actin[23]. Actin SiMS and xTalin1 SiMS flowed at a similar velocity (Fig. 1e, f, Supplementary Movie 2). Vinculin SiMS also flowed at a similar speed to the retrograde actin flow. xTalin1 and vinculin in FAs flowed at a similar speed to the side-flowing actin SiMS (Fig. 1g). For both xTalin1 and vinculin, there was no significant difference in the speckle speed ratio between molecules that flowed over FAs and out of FAs (Fig. 1g), indicating that thin FAs have little effect on the speeds of flowing FA component molecules. Therefore, our SiMS analysis suggests that flowing xTalin1 and vinculin SiMS associate with the lamellipodial actin network without being linked to the substrate.

We verified the uniform actin flow in lamellipodia containing FAs by using the nanometer-scale displacement measurement of actin SiMS[16] (Fig. S1). We previously reported that the actin SiMS approaching the FA area slow down within 0.5 μm of the outer edge of mature FAs, whereas the flow speeds of actin SiMS are uniform in the center of mature FAs[16]. We confirmed that all observed F-actin in FAs continuously moved at similar flow speeds measured with a low localization error of 11.7 nm with 500-ms temporal resolution (Fig. S1C). These results support that the F-actin network constantly moves as a single unit in the center of FAs.

We next measured the proportions of xTalin1 molecular states and the switching kinetics of xTalin1 SiMS between flowing and stationary states. The fraction of xTalin1 in a speckle state was calculated by dividing the number of the observed SiMS by the total number of EGFP-xTalin1 molecules measured from the integrated fluorescent intensity of EGFP in the observed region. In lamellipodia, about 19% of the EGFP-xTalin1 molecules were in a speckle state, whereas the other EGFP-xTalin1 molecules were in a diffusing state. Among EGFP-xTalin1 SiMS, 57% were in a stationary state and 39% were in a flowing state (Table S1, Fig. 1h). We measured the lifetime distributions of flowing xTalin1 SiMS and stationary xTalin1 SiMS with images acquired at 2-s intervals (Fig. S2). Of the stationary xTalin1 SiMS, about 87% dissociated from the substrate and transitioned to a diffusing state with a dissociation rate constant ($k_1$) of 0.170 s$^{-1}$, whereas 13% switched motions from stationary to flowing with a rate constant ($k_2$) of 0.0324 s$^{-1}$ (Fig. 1h). Of flowing xTalin1 SiMS, about 92% dissociated from F-actin and transitioned to a diffusing state with a dissociation rate constant ($k_3$) of 0.104 s$^{-1}$, whereas 8% switched motions from flowing to stationary with a rate constant ($k_4$) of 0.00694 s$^{-1}$ (Fig. 1h). These observations revealed a dynamic transition of xTalin1 between stationary, flowing and diffusing states in lamellipodia.

### Nanometer-scale transient clutch of Talin

Since filamentous actin continuously shifts ~5 subunits/sec with the retrograde actin flow (~30 nm/s) in lamellipodia, the linkage between F-actin and integrin via Talin could be transient (~0.2 s). Therefore, we increased the spatiotemporal resolution of displacement measurements of xTalin1 SiMS to analyze the switching process. We used unattenuated 75 W xenon illumination to monitor EGFP-tagged xTalin1 SiMS with a high signal-to-noise ratio, and acquired images at 100-ms intervals for the short duration of 10 s (Fig. 2a). Subpixel localization of the centroids of EGFP-tagged xTalin1 SiMS was determined with the two-dimensional Gaussian fit model of Speckle TrackerJ[16,24]. Under this condition, the centroids of immobile EGFP-xTalin1 SiMS on the glass surface were distributed with a SD of 18.6 nm and 18.7 nm in the *x*- and *y*-axes, respectively.

The nanometer-scale displacement analysis revealed two types of movement of EGFP-tagged xTalin1, indicative of actin flow-induced molecular stretching of Talin. First, when xTalin1 SiMS switched from stationary to flowing behavior, the centroid of EGFP fused at the N-terminus of xTalin1 jumped in the direction of the flow (Fig. 2b, left). By contrast, with EGFP fused at the C-terminus, a large displacement in the actin flow direction was scarcely observed upon switching from stationary to flowing behavior (Fig. 2b, right). We measured the distance ($\Delta x$) in the flow direction when the switching was first detected (see Methods). EGFP-xTalin1 SiMS exhibited significantly larger $\Delta x$ values than xTalin1-EGFP SiMS (Fig. 2c). For $\Delta x$ of EGFP-xTalin1, the greatest number of observed speckles showed displacement between 40 and 60 nm, and the $\Delta x$ value ranged up to 134 nm (Fig. 2d). We estimate that our measurement is reliable for xTalin1 SiMS jump 40 nm or more (Fig. S3). On the other hand, $\Delta x$ of xTalin1-EGFP was 31.1 nm on average (Fig. 2c). This is presumably due to the conformation change of Talin from a 15-nm globular to a ~60-nm open form upon F-actin binding[25].

Second, xTalin1-EGFP SiMS in a stationary state occasionally moved back and forth in the actin flow direction (Fig. 2e, f). Table S2 shows the classification of xTalin1 SiMS (lifetime ≥2 s) in the images acquired at 100 ms intervals. The nanometer-scale displacement analysis revealed that 6.54% of xTalin1-EGFP SiMS exhibited back-and-forth motions in a 10 s observation window (Table S2), which corresponds to 14% of stationary xTalin1-EGFP. By contrast, we did not observe such back-and-forth motion of EGFP-fused at the N-terminus of xTalin1 (EGFP-xTalin1) (Table S2).

These jumping and back-and-forth motions suggest there is a conformational change of xTalin1 associated with a transient linkage

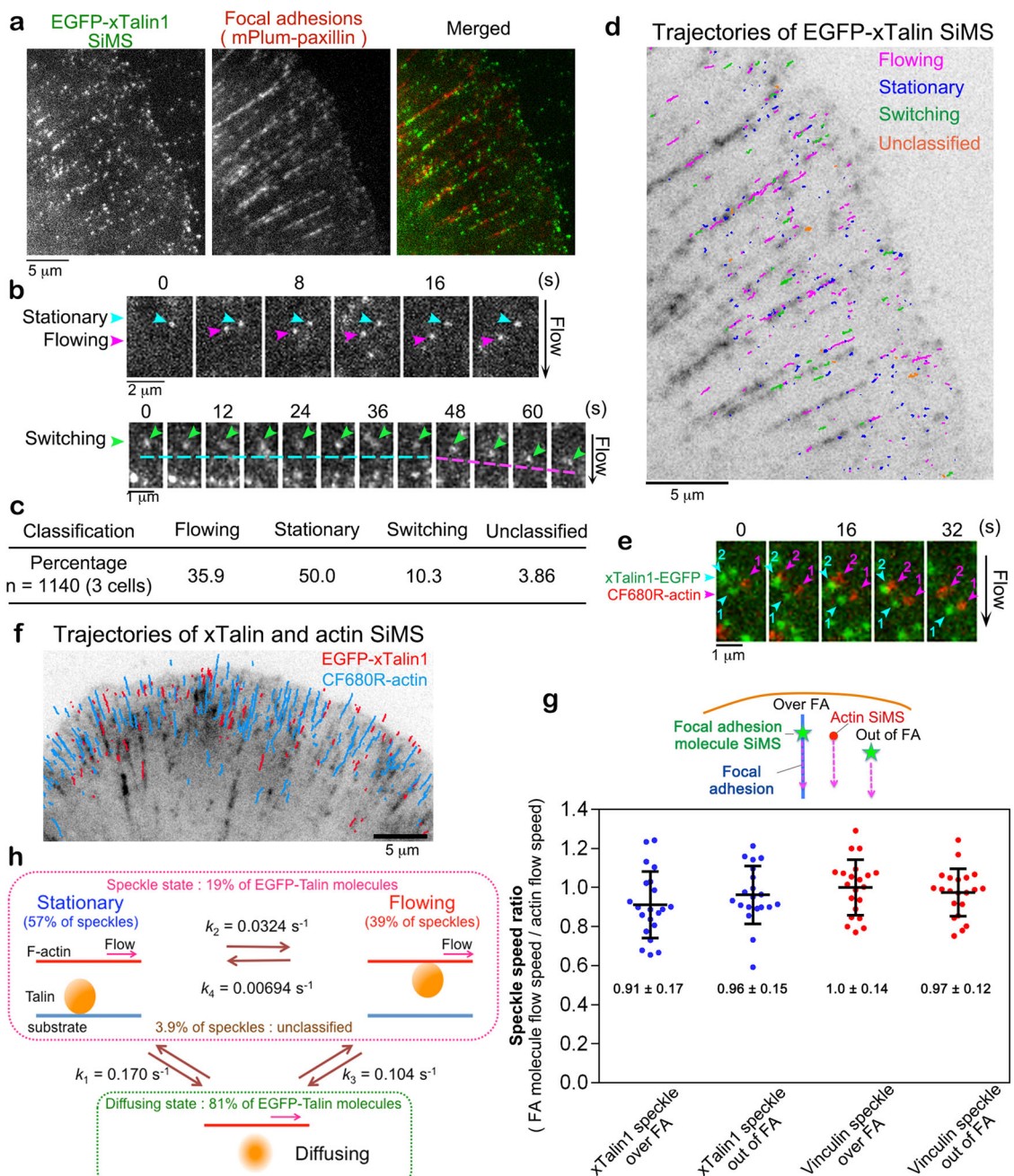

**Fig. 1 | Talin1 flows with a similar velocity to the retrograde actin flow in lamellipodia of XTC cells. a** Images of EGFP-xTalin1 SiMS (left) and mPlum-paxillin (middle) that marks FAs in lamellipodia. **b** The time-lapse images show representative motions of xTalin1 SiMS, which are stationary (upper, blue arrows), flowing (upper, pink arrows) and switching (lower, green arrows). The switching speckle in the lower row shows an example where the motion switches from stationary to flowing. **c** EGFP-xTalin1 SiMS found within a 120-s time window are classified into the indicated groups. **d** A trajectory map of xTalin1 SiMS observed within a 120-s time window is shown in an image of mPlum-paxillin (grey). Colors of lines indicate the indicated motions of speckles. **e** Merged time-lapse images of EGFP-xTalin1 SiMS (green) and CF680R-actin SiMS (red). The EGFP-xTalin1 SiMS with numbers (numbered blue arrowheads) and the CF680R-actin SiMS with numbers (numbered pink arrowheads) flow with a similar velocity. **f** Overlaid trajectories of EGFP-xTalin1 SiMS in a flowing motion (red lines) and CF680R-actin SiMS (blue lines) observed within a 120-s time window are shown in an image of

mPlum-paxillin (grey). **g** The speckle speed ratio of the flow speeds of xTalin1 SiMS or vinculin SiMS to those of side-flowing CF680R-actin SiMS. The speeds of xTalin1 SiMS ($n = 21$) and vinculin SiMS ($n = 21$) flowing over FAs or those of xTalin1 SiMS ($n = 21$) and vinculin SiMS ($n = 20$) flowing outside adhesions were divided by the speeds of actin SiMS that flowed near and outside adhesions (side-flow) as shown in the upper schematic diagram. Data are pooled from two independent experiments for all conditions. Bars and values indicate mean ± SD. **h** Summary of single-molecule kinetics of xTalin1 in lamellipodia. $k_1$ represents the rate constant for the dissociation of stationary EGFP-xTalin1 SiMS to diffusion. $k_2$ represents the rate constant for the transition of EGFP-xTalin1 SiMS from stationary to flowing motions. $k_3$ represents the rate constant for the dissociation of flowing EGFP-xTalin1 SiMS to diffusion. $k_4$ represents the rate constant for the transition of EGFP-xTalin1 SiMS from flowing to stationary motions. Source data are provided as a Source Data file.

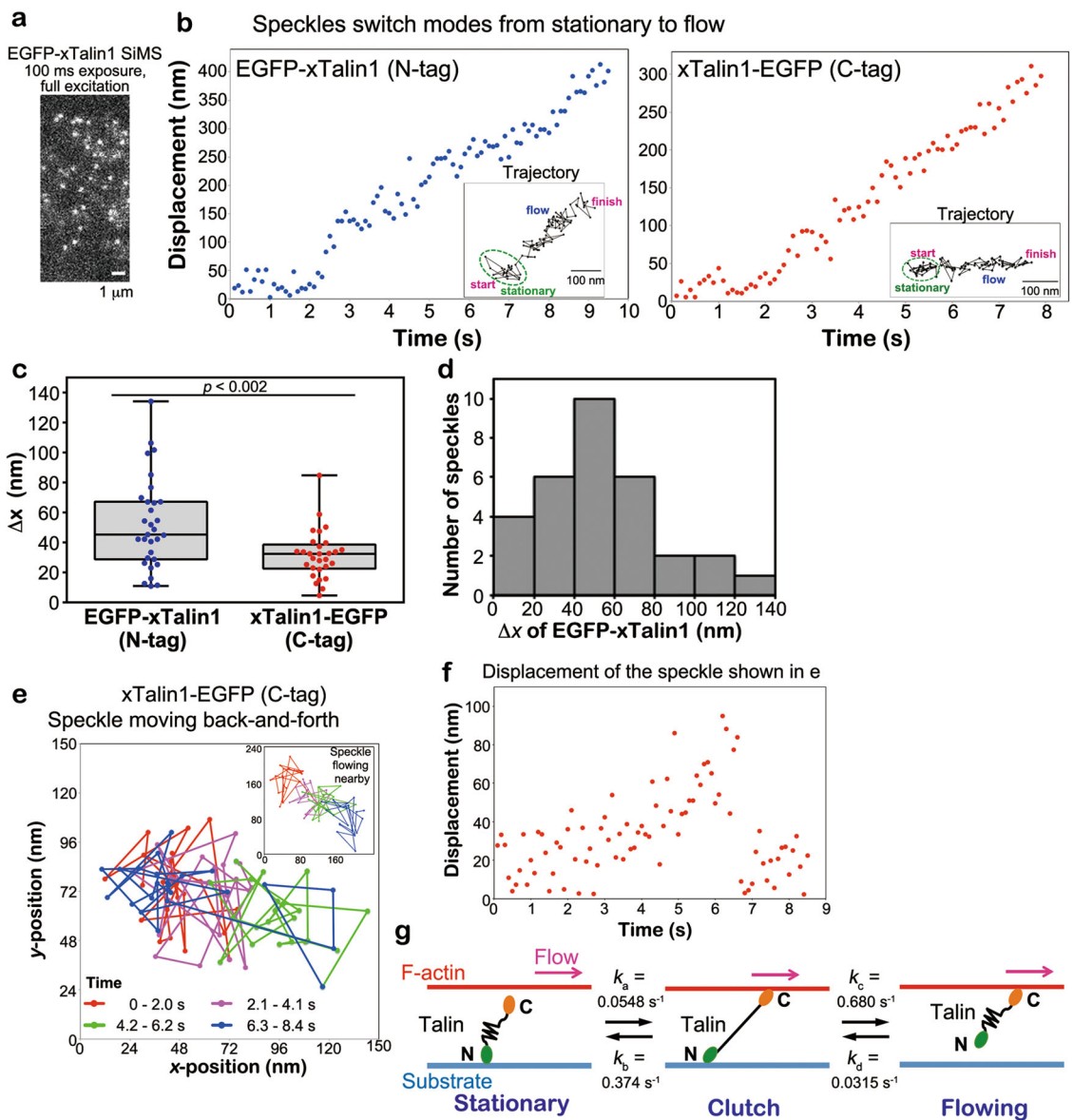

**Fig. 2 | Nanometer displacement measurements of EGFP-tagged xTalin1 SiMS.**
**a** An image of EGFP-xTalin1 SiMS acquired with 100-ms exposure time and unattenuated 75-W xenon illumination. **b** Representative displacement plots of the central position of EGFP-xTalin1 (N-tag, left) and xTalin1-EGFP (C-tag, right) in the series of fast-tracking images. The central position was determined by the Gaussian-Fit model of Speckle TrackerJ software. The EGFP-xTalin1 SiMS and the xTalin1-EGFP SiMS switch the motion from stationary to flowing. Insets show trajectories of the same xTalin1 SiMS. **c** Box-and-whisker plots of the distance (Δx) in the flow direction between the position at the stationary state and the first position where the flow is detected. Box-and-whisker plot shows the median (the line in the box), the 25th and 75th percentiles (box), and the minimum and maximum values (whiskers). Data represents Δx of $n = 31$ EGFP-xTalin1 SiMS (blue dots) from seven independent experiments and those of $n = 29$ xTalin1-EGFP SiMS (red dots) from 6 independent experiments. The p value (0.00190) was determined using a two-tailed t-test. In our estimation, xTalin1 SiMS that jumps 40 nm or more can be measured with more than 75% probability (Fig. S3). **d** The Δx distribution of EGFP-xTalin1 SiMS shown in (**c**). **e** A representative trajectory of the central position of xTalin1-EGFP SiMS that exhibits back-and-forth motion. The dots indicate the central position of the speckle for each frame and are connected by lines (85 frames). The dots and lines are color-coded by the time after the start of tracking, as indicated. Inset shows the trajectory of a speckle flowing nearby, shown in the same color-coding. **f** Displacement plot of the central position of xTalin1-EGFP SiMS shown in (**e**). **g** Summary of switching kinetics of xTalin1 engaging and disengaging via an elastic transient clutch in Talin. $k_a$ represents the rate constant for the transition of stationary xTalin1 SiMS to a clutch state. $k_b$ represents the rate constant for the transition of xTalin1 SiMS from clutching to a stationary state. $k_c$ represents the rate constant for transition of clutching xTalin1 SiMS to a flowing state. $k_d$ represents the rate constant for the transition of flowing xTalin1 SiMS to a stationary state. Source data are provided as a Source Data file.

between the flowing F-actin and integrin (Fig. 2g). Talin contains an N-terminal head domain that binds integrin cytoplasmic domains and a C-terminal rod domain that binds F-actin via two ABSs (Fig. 3c)[3,4,7]. The C-terminal rod domain is comprised of 13 α-helical rod subdomains. In vitro single-molecule studies have demonstrated that subdomains of the C-terminal rod domain unfold and refold in response to mechanical forces[6,9,26]. Our data suggest that in a stationary state, Talin is linked to the substrate via integrins bound to its

N-terminal head domain. When the C-terminal domain of Talin binds to F-actin, the force of the retrograde actin flow is exerted on Talin, causing unfolding of the C-terminal rod domain in a clutch state (Fig. 2g). Talin in a clutch state either dissociates from integrin and switches to a flowing state, or dissociates from the actin filament and returns to a stationary state (Fig. 2g).

We next calculated the kinetics of elastic transient clutch engagement and disengagement (Fig. 2g, see Methods). The stationary

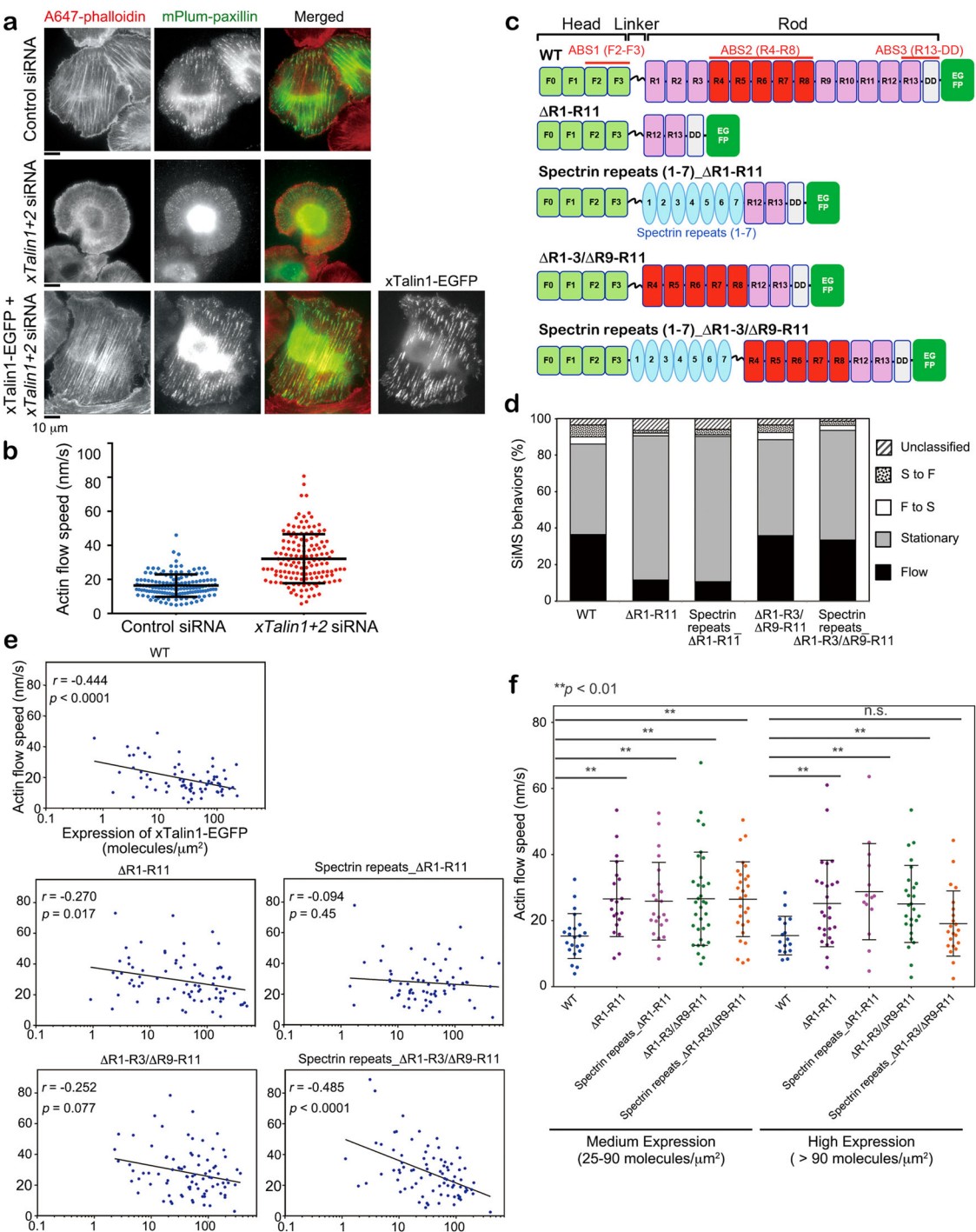

xTalin1 transition to a clutching state with a rate constant $k_a = 0.0548\ \text{s}^{-1}$, which was measured from time taken for stationary xTalin1-EGFP SiMS to switch to flowing or back-and-forth motion ($n = 129$). The clutching xTalin1 transition to a flowing state with a rate constant $k_c = 0.680\ \text{s}^{-1}$, which was approximated from the average clutch duration (1.47 s, $n = 31$). The clutching xTalin1 transition to a stationary state with a rate constant $k_b = 0.374\ \text{s}^{-1}$, which was determined from $k_c$ and the ratio of the frequency at which stationary xTalin1-EGFP switched to a flowing state to the frequency at which back-and-forth motion of xTalin1-EGFP observed (Table S2). We calculated the rate constant $k_d = 0.0315\ \text{s}^{-1}$ by using the rate constants $k_a$, $k_b$, and $k_c$, as described in Methods. Given these parameters, we estimated the proportion of xTalin1 SiMS in a clutch state was 4.1% (see Methods). These results revealed that only a small portion of xTalin1

molecules are mechanically linked and transmit forces accompanying unfolding of the Talin rod domain.

## Talin stretchability facilitates force transmission

We hypothesized that force-induced unfolding of α-helical bundle subdomains may inherently mediate force transmission at FAs, independent of the exposure of cryptic vinculin-binding sites in the Talin rod domain. To test this, we analyzed the functionality of a series of xTalin1 mutant proteins with distinct rod domains (Fig. 3c). We used siRNAs to deplete *Xenopus* Talin1 and Talin2, which suppressed expression of xTalins by 82.4% in *Xenopus* epithelial A6 cells (xTalin1/2 KD cells, Fig. S4). Depletion of both xTalin1 and xTalin2 caused a marked reduction in actin stress fibers and FAs (Fig. 3a, middle row) compared to cells treated with control siRNA (Fig. 3a,

**Fig. 3 | Elastic α-helical bundle structures of the spectrin-repeats can substitute for xTalin1 stretchable domains. a** Images of F-actin and mPlum-paxillin in control (upper) and *xtalin1* and *xtalin2* double-knockdown (middle) *Xenopus* A6 cells (xTalin1/2 KD cells). F-actin was stained with Alexa 647 phalloidin (A647-phalloidin, left). Images in the lower column show F-actin and mPlum-paxillin in an xTalin1/2 KD cell expressing xTalin1-EGFP. **b** Depletion of xTalin1 and xTalin2 attenuates the retrograde actin flow speeds in lamellipodia. The actin flow speeds in control ($n = 142$, blue dots) and xTalin1/2 KD ($n = 144$, red dots) cells are shown. Each dot represents the average speed of 10 actin speckles in individual cells. Data are pooled from 17 independent experiments for both conditions. Bars indicate mean ± SD. **c** Schematic diagram of xTalin1 constructs expressed as C-terminally tagged EGFP-fusion proteins. **d** SiMS behaviors of xTalin1 and mutant constructs expressed in xTalin1/2 KD cells. SiMS of xTalin1 and the respective xTalin1 mutants are classified into the indicated groups. The data represents the results of analyzing xTalin1WT in a total of 330 SiMS, xTalin1ΔR1-R11 in a total 313 SiMS, xTalin1 spectrin repeats (1-7)_ΔR1-R11 in a total 234 SiMS, xTalin1ΔR1-R3/ΔR9-R11 in a total 225 SiMS, and xTalin1 spectrin repeats (1–7)_ ΔR1-R3/ΔR9-R11 in a total 410 SiMS. The actin-binding site 2 (ABS2) of xTalin1 is required for binding to F-actin in lamellipodia. **e** Relationship between the retrograde actin flow speeds and the expression level of EGFP-tagged xTalin1 and respective mutants. The data represents the results of analyzing xTalin1/2 KD cells expression xTalin1WT ($n = 72$ cells), xTalin1ΔR1-R11 ($n = 78$), xTalin1 spectrin repeats (1–7)_ΔR1-R11 ($n = 68$), xTalin1ΔR1-R3/ΔR9-R11 ($n = 86$), and xTalin1 spectrin repeats (1-7)_ ΔR1-R3/ΔR9-R11 ($n = 83$). Pearson's correlation coefficients ($r$) and $p$-values (two-tailed) are indicated. Data are pooled from more than six independent experiments for all conditions **f**. The retrograde actin flow speeds in xTalin1/2 KD cells expressing xTalin1 (WT) and the indicated xTalin1 mutants at medium (med; 25–90 molecules /μm²) and high (> 90 molecules/μm²) levels. Bars indicate mean ± SD. **, $p < 0.01$, one-tailed Student's $t$ test. The exact $p$ values are provided in Source Data. The results in e and f show that the xTalin1ΔR1-R3/ΔR9-R11 construct did not rescue a decrease in the retrograde actin flow speeds, but the xTalin1 mutant with spectrin repeats (1-7)_ΔR1-R3/ΔR9-R11, did rescue retrograde actin flow speeds at high expression levels. The data represents the results of analyzing xTalin1/2 KD cells expression xTalin1WT ($n = 28$ for med and $n = 16$ for high), xTalin1ΔR1-R11 ($n = 24$ for med and $n = 26$ for high), xTalin1 spectrin repeats (1-7)_ΔR1-R11 ($n = 26$ for med and $n = 14$ for high), xTalin1ΔR1-R3/ΔR9-R11 ($n = 33$ for med and $n = 26$ for high), and xTalin1 spectrin repeats (1-7)_ ΔR1-R3/ΔR9-R11 ($n = 32$ for med and $n = 22$ for high). Source data are provided as a Source Data file.

upper row). Expression of wild-type (WT) xTalin1-EGFP rescued the *xTalin1* and *xTalin2* knockdown phenotype (Fig. 3a, lower row). SiMS analysis of actin revealed that the retrograde actin flow speeds in lamellipodia were markedly faster in the xTalin1/2 KD cells than the cells treated with control siRNA (Fig. 3b). These observations are consistent with the previous studies of Talin-deficient cells[10,27].

We next investigated the role of the Talin rod domain in the force transmission between the actin network and the substrate by monitoring the speed of the retrograde actin flow. We first verified the actin-binding properties of xTalin1 mutant proteins. EGFP-xTalin1 WT exhibited similar SiMS behaviors in the xTalin1/2 KD A6 cells (Fig. 3d) to those observed in XTC cells (Fig. 1c). Deletion of subdomains R1-R11 reduced the percentage of flowing SiMS to about 25% of xTalin1 WT (Fig. 3c, d). To confirm the contribution of xTalin1 ABS2 to F-actin binding in lamellipodia, we tested the xTalin1 mutant ΔR1-R3/ΔR9-R11, in which ABS2 consisting of R4-R8 was added to ΔR1-R11 (Fig. 3c). The percentage of ΔR1-R3/ΔR9-R11 SiMS in a flowing state was restored to the same level as xTalin1 WT (Fig. 3d). These results suggest that ABS2 contributes to the binding of xTalin1 to the flowing actin network.

With regard to the effect on the retrograde actin flow, expression of xTalin1 WT-EGFP decreased the flow speed in xTalin1/2 KD cells depending on its expression level (Fig. 3e). The higher the expression level of xTalin1 WT was, the more the actin flow speed was suppressed (Fig. 3e). Expression of xTalin1 ΔR1-R11 did not significantly affect the retrograde flow speeds in the xTalin1/2 KD cells (Fig. 3e) even at high expression levels (Fig. 3f). Notably, overexpression of xTalin1 ΔR1-R3/ΔR9-R11 had a much weaker effect on the retrograde flow speed in xTalin1/2 KD cells than that of xTalin1 WT (Fig. 3f).

To investigate the contribution of unfolding of α-helical bundle subdomains to the force transmission, we designed two chimeric mutants in which α-helical β-spectrin repeats 1 to 7 were introduced into xTalin1ΔR1-R11 and ΔR1-R3/ΔR9-R11 mutants (Fig. 3c, hereafter designated as xTalin1 SR_ΔR1-R11 and xTalin1 SR_ΔR1-R3/ΔR9-R11, respectively). Spectrin repeats are α-helical 3-helix bundle domains that can be unfolded by forces in the range of 10 pN[28]. Similar to ΔR1-R11, xTalin1 SR_ΔR1-R11 lacking ABS2 showed a smaller fraction of flowing SiMS than xTalin1 WT (Fig. 3d). Expression of xTalin1 SR_ΔR1-R11 did not significantly affect the retrograde flow speeds in the xTalin1/2 KD cells (Fig. 3e, f).

In contrast, the percentage of xTalin1 SR_ΔR1-R3/ΔR9-R11, containing ABS2, in a flowing state was almost the same as xTalin1 WT and ΔR1-R3/ΔR9-R11 (Fig. 3d), suggesting that xTalin1 ΔR1-R3/ΔR9-R11 and SR_ΔR1-R3/ΔR9-R11 mutants interact with F-actin at a similar frequency. Expression of xTalin1 SR_ΔR1-R3/ΔR9-R11 remarkably reduced the retrograde actin flow speed in xTalin1/2 KD cells at high expression levels (Fig. 3e, f). A higher correlation coefficient was found between the expression level and the retrograde actin flow speed in cells expressing xTalin1 SR_ΔR1-R3/ΔR9-R11 than xTalin1 mutants lacking either spectrin repeats or ABS2 (Fig. 3e). These results suggest that unfolding of spectrin repeats is promoted by the interaction of ABS2 with the flowing actin network, and the stretchable property of Talin is important for force transmission between F-actin and the substrate.

## Talin unfolding increases clutch duration and work

To quantify how the number and unfolding properties of Talin rod subdomains transmit force between the flowing F-actin network and the substrate, we used a coarse-grained mathematical model that represents the Talin rod subdomains as bead and spring segments (Fig. 4a, see Methods). We assumed that Talin contains $N$ folded, stiff rod subdomains, and two additional subdomains binding to integrin and F-actin on each end, respectively (Fig. 4a). The actin-associated bead was assumed to move at the rate of retrograde flow while the integrin-associated bead remained stationary, leading to an increase in tension along Talin over time. Talin subdomains were assumed to unfold according to Bell's law, using parameters corresponding to the most common measured values[9]. Unfolded Talin domains were modeled as freely jointed chains. We assumed that Talin dissociation from integrin follows Bell's law with $k_{unbind}(F) = k_{unbind,0} \exp(\Delta x_{unbind} F / kT)$, estimating the unbinding rate at zero force $F$, $k_{unbind,0} = 0.17 s^{-1}$ from our SiMS data (Fig. 1g). For simplicity, we do not consider the less-frequent case of dissociation of Talin from actin. We estimated $\Delta x_{unbind} = 0.51 \, nm$ as the value that best reproduced the experimental unbinding displacement distributions in Fig. 2d (Methods).

By numerically solving the master equation of the system, based on force balance as individual subdomains unfold stochastically, as well as with explicit stochastic simulations (Supplementary Movie 3), we calculated the average clutch duration and the average work done by Talin on actin per linkage. These distributions and their averages depend on the initial angle $\theta$ of Talin with respect to the substrate when it initiates the linkage between integrin and F-actin (Fig. 4a). For both vertical, $\theta = 90°$, and $\theta = 45°$ cases, and keeping the total number of subdomains at $N = 12$, the average clutch duration and work done on actin increased rapidly with the number of subdomains that can unfold up to about five, above which it increased at a smaller rate (Fig. 4b, c). Therefore, our model showed that mechanically unfolding subdomains have the effect of improving the force transmission associated with the clutch. We also simulated the effect of changing $N$, the total number of unfolding Talin subdomains, for a chain at $\theta = 45°$ and found that the work done peaks at $N = 3–13$, followed by a slow decline, as the retrograde flow speeds vary from 10 to 40 nm/s (Fig. 4d). The trends in Fig. 4d depend on the value of $\Delta x_{unbind}$, with weaker Talin-integrin binding (larger $\Delta x_{unbind}$) having an optimum work at larger $N$

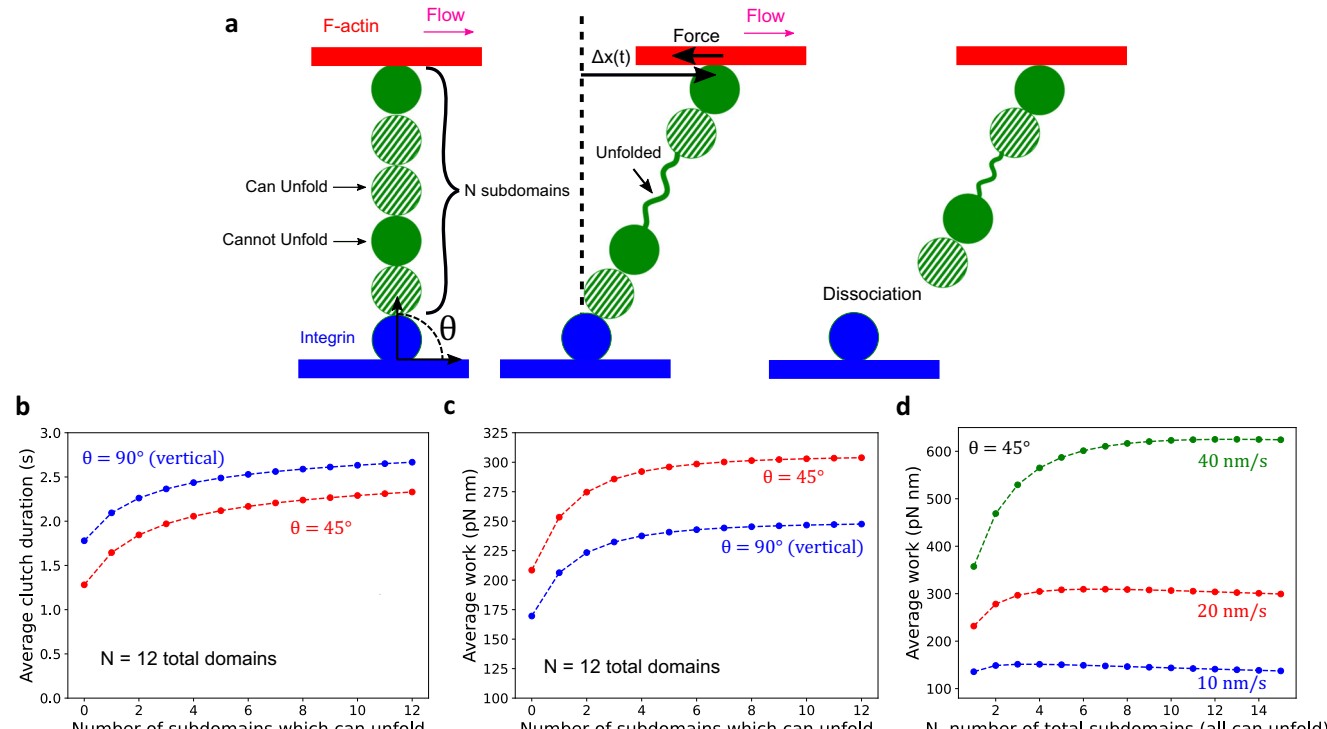

**Fig. 4 | Mathematical model of a single Talin chain. a** Talin is represented as a string of rod subdomains (green), the ends of which associate with stationary integrin (blue) and permanently with F-actin (red) moving at constant retrograde flow speed. The model is studied as a function of the total number of rod sub-domains that can unfold (dashed green) or not (solid green). We varied the initial angle $\theta$ between the axis of Talin, assumed that it started linearly in the plane of the figure. The master equation of the system evolves over time, satisfying the force balance along Talin, up to the point of dissociation from integrin. The work on F-actin is calculated as the sum of the product of the force on actin along the flow direction times the displacement. **b** Average clutch duration as a function of the number of subdomains that can unfold, with the total subdomain number fixed at

12 and retrograde flow speed 20 nm/s. The clutch duration is longer for chains that are at $\theta = 90°$ compared to 45° since Talin tension increases faster at lower values of $\theta$. **c** Average work as function of the number of subdomains that can unfold, with the total subdomain number fixed at 12. The work is less at $\theta = 90°$ compared to 45°, since at a starting angle of $\theta = 90°$, it takes longer for Talin to be fully stretched, and integrin dissociation may meanwhile occur. **d** Average work as a function of the total number of rod subdomains, all assumed to be able to unfold, and retrograde flow speed. (In this calculation we assume a fixed retrograde speed and do not consider the feedback between retrograde flow and clutch strength in the whole lamellipodium, which would associate lower retrograde flow speeds with higher work done by Talin).

(Fig. S5). Furthermore, in order to investigate the impact of model assumptions, we performed three additional simulations, of which results showed that the unfolding of Talin rod subdomains promotes force transmission associated with the clutch (Fig. S8, see Methods). Overall, our model suggests that efficient force transmission within the range of cellular retrograde flow speeds depends on the unfolding properties of Talin subdomains, their total number, as well as the strength of the integrin-Talin and Talin-actin bonds.

## Discussion

The present study has elucidated that force transmission requires the elastic transient clutch of Talin accompanying unfolding of the Talin rod subdomains. Talin is essential for tethering the flowing F-actin network to the immobile substrate[10]. On thin elongated FAs in lamel-lipodia, F-actin flows uniformly at 20–50 nm/s[16]. F-actin also con-tinuously move backward at 10–20 nm/s on mature FAs[16] (Fig. S1). Thus F-actin constantly shifts 2–9 subunits/s along with the retrograde flow on thin and mature FAs. The stretchable property of the Talin rod subdomains thus appears to be a prerequisite to link the moving F-actin to the substrate. We indeed observed that the increased actin flow speed in Talin knockdown cells is restored by expression of a Talin mutant, in which the Talin rod subdomains are replaced with the stretchable repeats of β-spectrin. This observation indicates that the stretchable property is critical for force transmission, in addition to the exposure of cryptic binding sites in the Talin rod domain[12].

Talin has been extensively studied for its force-induced unfolding property by in vitro single-molecule experiments[6,9,26]. The unfolding of

Talin rod subdomains has been proposed to act as a mechanosensor to control force-dependent interactions with its binding proteins[5,6,26]. The unfolding of Talin rod domain may also act as a force buffer that maintains mechanical equilibrium at low forces over a wide range of Talin extensions[9]. The Talin extension fluctuations in response to changes in force have been suggested to be responsible for Talin's mechanosensing functions[9]. However, the lifetime and kinetics of the linkage by Talin in cells were unknown, and the role of the unfolding of Talin in force transmission has not been critically examined in the previous studies[6,9,26]. Owing to the capability of the SiMS approach, our present study directly visualized stochastic, transient linkage of Talin in live cells. Furthermore, our simulations provide evidence that sto-chastic unfolding of subdomains facilitates the force transmission between the flowing F-actin network and the substrate by extending the clutch duration. Our study thus elucidates the direct link between molecular biophysics and cellular mechanics.

We estimated how much force is transmitted per single xTalin1 molecule in XTC cells. The concentration of Talin1 in cultured cell lines is 3–5 μM[29]. In XTC cells, about 6.4% of EGFP-xTalin1 molecules are in a speckle state, whereas the other molecules are diffusing in the cyto-plasm. We previously measured that the cytosol volume of a XTC cell is 0.68 pL[30]. From these values, we estimate the number of xTalin1 molecules in a XTC cell to be $1.2 \times 10^6$–$2.1 \times 10^6$, therefore the number of xTalin1 in a speckle state is $8 \times 10^4$–$1.3 \times 10^5$ and the number of xTalin1 in a clutch state (4.1% of SiMS) is 3300–5400. The force of actin polymerization at lamellipodial tip is one of the main driving forces of the retrograde flow[31]. The flow speed decreases from 50 nm/s in the

Talin knockdown cells in which the effects of Talin depletion are pronounced to 18 nm/s in control cells (Fig. 3b). Therefore we estimate the force to restrict the retrograde flow by clutch of Talin per actin filament to be ≈1.8 pN[32] The number of filaments in contact with 10 μm-wide lamellipodium edge (roughly 1/16 of the cell edge) of a XTC cell is estimated to be ≈2500[32], therefore the sum of the force exerted on Talin is 4500 pN. If the force is applied equally to all Talin SiMS in the area, it is only 0.55–0.9 pN per molecule, which is much lower than the force that induces unfolding of the Talin rod subdomains. Because only 4.1% of Talin SiMS are mechanically engaged, the force applied to a single Talin in a clutch state is estimated to be 13–22 pN. This range of forces induces unfolding of the Talin rod subdomains in vitro[9,26] and corresponds to forces transmitted to individual integrins in living cells[33]. Therefore, the elastic transient clutch of Talin quantitatively explains force transmission at FAs in living cells. The majority and excess of non-linked Talin may be important to compensate for dynamic linkage between moving F-actin and integrins. Our kinetics data indicate that without this population, the linkage at FAs cannot be maintained.

Our simulations suggest that unfolding of the Talin rod subdomains prolongs the clutch duration, thereby increasing force transmission at FAs. Unfolding of even a single subdomain leads to a substantial increase in the clutch duration and work. Therefore, the F-actin flow-dependent force produces work at FAs by stochastic coupling and repetitive stretch of Talin. Interestingly, the unfolding properties of Talin rod subdomains in vitro[9] are close to the optimal values for maximum work in our simulation (Fig. S6). The in vitro study also shows that 5 out of 12 Talin rod subdomains are unfolded at <15 pN[9]. Our simulations predict that the work done increases with the number of subdomains which can unfold, up to about five. We speculate that the force-induced extensibility and the number of Talin rod subdomains might have been optimized for force transmission during the evolution of Talin.

The elastic transient clutch of Talin shares similarities with the sacrificial bond and hidden length system that is a mechanism that increases the toughness of biological materials including bone and shells[34–38]. Sacrificial bonds are defined as additional bonds that break before the main structural link is broken[35]. The system prevents materials from fracturing by enhancing energy dissipation[34,36]. Similarly, the Talin rod domain unfolds to maintain the link between F-actin and integrins for prolonged durations. However, the elastic transient clutch of Talin conceptually differs from the sacrificial bond, because unfolding of the Talin rod domain serves as a force transducer at FAs, increasing the work per molecule by sustaining individual connections between integrin and actin for longer times. The function of unfolding of the Talin rod domain as a force transducer is also different from the proposed role as a force buffer to maintain mechanical equilibrium at low forces[9].

Our SiMS data show that the behaviors of Talin molecules are clearly divided into two modes, flowing and stationary, which has not been reported in the previous quantitative imaging of Talin[14,39,40]. In the previous studies, the quantification may have failed to detect the two modes due to measuring the mixed population[14,39,40]. In analyses of tracking particles, the accuracy of tracking decreases drastically at high signal density[41]. In this study, we took particular care to accurately track individual Talin SiMS.

In conclusion, this study has elucidated a role for protein unfolding in force transmission between the moving F-actin network and immobile adhesion structures. In the macroscopic world, machines transmit forces through rigid joints and structures. On the other hand, in the microscopic world, nanometer-scale molecules must connect micrometer-scale structures that move at different speeds. Molecular stretching may be a fundamental mechanism for force transmission as exemplified by the elastic transient clutch of Talin.

## Methods

### Plasmids and fluorescent dye-labeled actin

Two *Xenopus laevis* Talin1 cDNA clones (GenBank accession numbers: CF282569.1 and EB470267.1), which contain 5′- and 3′-cDNA fragments of *Xenopus* Talin1, respectively, were obtained from Open Biosystems. The two *Xenopus* Talin1 cDNA fragments were overlapped and fused to generate the full-length *Xenopus* Talin1 cDNA by using PCR. The full-length *Xenopus* Talin1 cDNA was subcloned into the expression vectors harboring delCMV[42] for EGFP-xTalin1 and xTalin1-EGFP. The expression plasmid for EGFP-tagged xTalin1 ΔR1-R11 (Δ482-2125) was constructed by inverse PCR, in which Pvu I site (5′-GGCCGATCGGGT-3′) was added to the end of primers for the subsequent ligation. To generate the expression plasmid for EGFP-tagged xTalin1 ΔR1-R3/ΔR9-R11 (Δ482-911 and Δ1658-2125), the sequence encoding xTalin1 R4-R8 (912-1657) was amplified by PCR and then inserted into the EGFP-fused xTalin1 ΔR1-R11 construct at the Pvu I site. Human ß2 spectrin (GenBank accession number: NG029817.1) was obtained from Addgene. To generate the expression plasmids for EGFP-tagged xTalin1 SR_ΔR1-R11 and xTalin1 SR_ΔR1-R3/ΔR9-R11, the sequence encoding human β2 spectrin repeats 1 to 7 (270-1041) was amplified by PCR, and then inserted between xTalin1 residues 417 and 418, which are within the linker region (401–481) of xTalin1 ΔR1-R11 and xTalin1 ΔR1-R3/ΔR9-R11. *Xenopus* paxillin cDNA (GenBank accession number: BC070716) was obtained from Open Biosystems and was subcloned into the mPlum expression vector harboring delCMV. The expression vector for EGFP human vinculin was described previously[16].

Rabbit skeletal muscle actin labeled with DyLight550-NHS ester (Thermo Fischer Scientific) and CF680R-NHS ester (Biotium) were prepared as described previously[16,23,43].

### Cell culture and electroporation of fluorescent-labeled actin

*Xenopus laevis* XTC cells were obtained and maintained as described previously[42,44]. *Xenopus laevis* A6 kidney epithelial cells (generously provided by Dr. Yuko Mimori-Kiyosue, RIKEN Center for Life Science Technologies) were maintained as described previously[45]. XTC cells and A6 cells were subjected to electroporation with the Neon transfection system (Invitrogen, Carlsbad, CA) to deliver DyLight550-labeled actin or CF680R-labeled actin as described previously[16,43].

### Single-molecule speckle (SiMS) imaging and data analysis

SiMS imaging in XTC cells and A6 cells were carried out as previously described[23,42–44]. Briefly, XTC cells were allowed to spread on a coverslip coated with 0.1 mg/ml poly-L-lysine (PLL) and 10 μg/ml laminin (Sigma-Aldrich) in 70% L15 Leibovitz medium (Invitrogen) without serum. A6 cells were allowed to spread on a coverslip coated with 0.1 mg/ml PLL and 0.1 mg/ml collagen (Nitta Gelatin) in 50% L15 Leibovitz medium without serum. Imaging was performed using a microscope (IX83, Olympus) equipped with 75-W xenon illumination and a cooled EMCCD camera (Evolve 512, Photometrics), or a microscope (IX71, Olympus) equipped with 100-W mercury illumination and a cooled EMCCD camera (iXon Ultra 894, Andor). Images were acquired using the MetaMorph software (Molecular Devices).

To analyze SiMS of EGFP-tagged proteins and actin, we used Speckle TrackerJ plug-in[24] of ImageJ as previously described[16,43]. The central position of SiMS was determined by the Gaussian-Fit model of Speckle TrackerJ plug-in.

### Flow speed measurement of F-actin, xTalin1, and vinculin

To measure the flow speed of xTalin1 and vinculin SiMS flowing over FAs or outside adhesions, FAs were visualized by mPlum-paxillin. Three color imaging for xTalin1-EGFP or EGFP-vinculin, mPlum-paxillin and CF680R-actin was performed in the same microscopic field of view. The flow speeds of xTalin1, vinculin, and actin SiMS were measured by the linear fit of centroid displacement with more than 4 frames (≥8 s).

## The classification of EGFP-tagged xTalin1 SiMS

EGFP-tagged xTalin1 and its mutants SiMS were classified according to their behavior as follows. For the classification of xTalin1 SiMS (lifetime ≥4 s) in the images acquired at 2 s intervals in a 120 s time window (Fig. 1c and Table S1), flowing speckles are the fraction showing directional motion in the retrograde actin flow for more than 2 sequential images. Stationary speckles are the fraction that stops the motion for more than two sequential images. Switching speckles are the fraction that switches behavior from flowing or stationary in two or more sequential images to the other motion in two or more sequential images. Unclassified speckles are the fraction, which did not fall into the three categories. For the classification of xTalin1 SiMS (lifetime ≥2 s) in the images acquired at 100 ms intervals in a 10 s time window (Table S2), flowing speckles are the fraction showing directional motion in the retrograde actin flow for more than ten sequential images. Stationary speckles are the fraction that stops the motion in for more than 10 sequential images. Switching speckles are the fraction that switches behavior from flowing or stationary in more than ten sequential images to the other motion in more than ten sequential images. The back-and-forth speckles are the fraction showing directional motion in the retrograde actin flow more than eight sequential images and then moving in the opposite direction of flow and stationary more than three sequential images. Unclassified speckles are the fraction, which did not fall into four categories.

## Kinetics measurement of xTalin1 SiMS between flowing, stationary, and diffusing states

The kinetics of xTalin1 SiMS shown in Fig. 1h were estimated by Kaplan-Meier survival analysis in GraphPad Prism (6.07). The lifetime of stationary or flowing xTalin1 SiMS (Fig. S2) and the time taken for switching xTalin1 SiMS to transition to the other motion were measured in the images acquired at 2 s intervals with a 120-s time window.

To analyze the switching kinetics from stationary xTalin SiMS to either a diffusing or a flowing state, we performed Kaplan-Meier survival analysis. We first set stationary xTalin1 SiMS ($n = 570$) that disappeared due to switching to a diffusing state or photobleaching as the event of interest. The event of stationary xTalin1 SiMS ($n = 83$) switching their motion to flowing was censored. Then, the observed rate of xTalin1 SiMS disappearance was calculated by fitting a single exponential to the survival curve. To calculate the rate constant ($k_1$) from stationary xTalin1 SiMS to a diffusing state, the observed rate of xTalin1 SiMS disappearance was normalized for photobleaching as described previously[42,44]. Briefly, to measure the photobleaching rate of EGFP, time-lapse images of bulk EGFP-xTalin1 fluorescence in a live cell were obtained by illuminating its entire cell body under the same illumination condition as in SiMS imaging. The average fluorescence intensity in an area of the cell was corrected by the subtracting background value measured in an area outside the cell on each image. The photobleaching rate of the fluorescence intensity was calculated by fitting a single exponential decay curve, which was given by $y = a*\exp(-0.0071x)$, where $x$, $y$ and $a$ are number of frames, the fluorescence intensity, and the initial fluorescence intensity, respectively. Photobleaching was normalized by subtracting the photobleaching rate from the observed disappearance rate, which gives the rate constant ($k_1$).

To calculate the rate constant ($k_2$) of stationary xTalin1 SiMS to a flowing state, we performed Kaplan-Meier survival analysis with stationary xTalin1 SiMS switching their motion to flowing as the event of interest, and stationary xTalin1 SiMS that disappeared was censored. The rate constant of stationary xTalin1 SiMS to switch their motion to flowing ($k_2$) was calculated by fitting a single exponential to the survival curve.

Similarly, the dissociation rate constant ($k_3$) of xTalin1 SiMS that dissociated from F-actin and transitioned to a diffusing state and the rate constant ($k_4$) of flowing xTalin1 SiMS to switch their motion to stationary were calculated with the lifetime of flowing xTalin1 SiMS ($n = 409$) and the time taken for flowing xTalin1 SiMS ($n = 33$) to switch their motion to stationary.

## Nanometer-scale displacement analysis of xTalin1 SiMS

The nanometer-scale displacement analysis of xTalin1 SiMS was performed as described previously[16,43]. EGFP-tagged xTalin1 SiMS were acquired with a 100 ms exposure time and a full 75 W xenon excitation (100 frames). The xTalin1 SiMS distance ($\Delta x$) in the flow direction when the switching was first detected was measured from the central position of xTalin1 SiMS in the series of images. First, the respective average values of the $x$- and $y$-coordinates of the central positions of the xTalin1 SiMS in a stationary state for more than eight frames were set as the origin of the coordinate axes. Next, the distance moved in the flow direction from the origin (the stationary position) for each frame was plotted against time. The time when the distance exceeds the localization error (18.6 nm) three times continuously in the flow direction was defined as a switching to a flowing motion. The first frame of the three consecutive frames was defined as the time when flow was first detected. Displacement along flow was calculated from a linear fit for the central positions of xTalin1 SiMS in a flowing mode, and the distance ($\Delta x$) was calculated by introducing the time when the flow was first detected. In the measurement of $\Delta x$ based on the criteria for xTalin1 SiMS switching from stationary to flowing described above, the sensitivity of the measurement depends on the actual jumping distance of xTalin1 SiMS (Fig. S3). We estimate that xTalin1 SiMS that jumps 40 nm or more can be measured with more than 75% probability (Fig. S3).

## Calculations of switching kinetics of xTalin1 SiMS between stationary, clutching, and flowing states

The switching kinetics of xTalin1 SiMS shown in Fig. 2G were calculated based on xTalin1 SiMS nanometer-scale displacement analysis of the series of images acquired at 100 ms intervals with a 10-s time window. Flux balance at steady state gives the xTalin1 SiMS proportions in stationary, clutching, and flowing states by the following equations:

$$\frac{dS}{dt} = -k_a S + k_b C + X = 0 \tag{1}$$

$$\frac{dC}{dt} = k_a S + k_d F - (k_b + k_c)C = 0 \tag{2}$$

$$\frac{dF}{dt} = k_c C - k_d F - X = 0 \tag{3}$$

$$X = k_2 S - k_4 F \tag{4}$$

Here, $S$, $C$, and $F$, respectively, represent proportions of stationary, clutching, and flowing xTalin1. The stationary xTalin1 and the flowing xTalin1 transition to a clutching state with rate constants $k_a$ and $k_d$, respectively. The clutching xTalin1 transition to either a stationary state or a flowing state with rate constants $k_b$ and $k_c$, respectively. $X$ represents the net flux of xTalin1 per second from the flowing fraction to the stationary fraction through the diffusing fraction. The stationary xTalin1 transitions to a flowing state with a rate constant $k_2$ and the flowing xTalin1 transitions to a stationary state with a rate constant $k_4$ (Fig. 1h).

The stationary xTalin1 transition to a clutching state with a rate constant $k_a = 0.0548\,s^{-1}$, which was measured from time taken for stationary xTalin1-EGFP SiMS to switch to flowing or back-and-forth motion ($n = 129$). The clutching xTalin1 transition to a flowing state with a rate constant $k_c = 0.680\,s^{-1}$, which was approximated by the reciprocal of the average clutch duration of EGFP-xTalin1 SiMS, as the

average frequency of switching from the clutching xTalin1 to a flowing state occurs per second. The average clutch duration (1.47 s, $n = 31$) was calculated by dividing the distance between the position at the stationary state and the first position where the flow is detected ($\Delta x$ in Fig. 2d) by the flow speed for each EGFP-xTalin1 SiMS. The clutching xTalin1 transition to a stationary state with a rate constant $k_b = 0.374\,s^{-1}$, which was approximated from $k_c$ and the ratio of the frequency at which stationary xTalin1-EGFP switched to a flowing state (marked with an asterisk in Table S2) and the frequency at which back-and-forth motion of xTalin1-EGFP observed (marked with a dagger in Table S2). Consistent with these xTalin1 proportion measurements, using $S = 57\%$ (Fig. 1h) and the calculate rate constant values in Eqs. (1)−(4) we find $F = 38.3\%$ and $C = 4.1\%$.

## RNA interference

As we reported previously[46], the retrograde flow speed is variable in lamellipodia of XTC cells for each cell, which made it difficult to analyze the effect of the *xTalin1* and *xTalin2* knockdown. In *Xenopus* epithelial A6 cells, the *xTalin1* and *xTalin2* siRNAs effectively suppress the expression of xTalins (Fig. S3). The retrograde flow speeds were significantly faster in the xTalin1/2KD A6 cells than the control siRNA-treated cells (Fig. 3b). For these reasons, we used A6 cells for analyzing the restoration of the flow speeds by expression of the Talin rod domain mutants in the xTalin-depleted cells.

siRNA-mediated knockdown of *Xenopus* Talin1 and Talin2 in A6 cells were performed by using custom Silencer Select siRNAs (Ambion). We used four siRNAs against non-coding region of xTalin1 (oligo sequences: 5′-CCUUGUUACCUCUUAUUAAtt-3′, 5′-GUGUAUAAA UGAGUAACAAtt-3′, 5′-GUCUGUGCCUUAUUCCUAUtt-3′, 5′-GCCUUUA AGUGCCUGAAUAtt-3′) and three siRNAs against coding region of xTalin2 (oligo sequences: 5′-GAGCAGACUGGAACCCUUAtt-3′, 5′-AGA UGAUUUUGGUAGAUGAtt-3′, 5′-GAAUUGGAAUCACAAAUUAtt-3′). A mixture of siRNAs (100 nM each) against xTalin1 and xTalin2 or Silencer® Select Negative Control #1 (700 nM) were transfected by electroporation with the Neon transfection system. Cells were incubated for 48 h, and then the transfection of siRNAs was repeated. The expression constructs were co-transfected with siRNAs at the first transfection, and DyLight550-labeled actin or CF680R-labeled actin was co-transfected with siRNAs at the second transfection. Forty-eight hours after the second transfection, cells were trypsinized, and seeded on PLL and collagen-coated coverslips for observation.

## Western blotting

For western blot analysis of A6 cells transfected with siRNAs, cells were lysed in lysis buffer (50 mM Tris-HCl pH7.4, 150 mM NaCl, 1% TritonX-100, 0.5% sodium deoxycholate, 0.1% SDS, 1 mM EDTA, 1 mM DTT and protease inhibitor cocktail) 48 h after the second transfection. The lysates were sonicated on ice, centrifuged at 20,000 × g for 15 min, and then Laemmli buffer was added to the supernatant. The samples were heated at 98 °C for 5 min, and then subjected to SDS-PAGE with precast 4-15% Tris-HCl gradient SDS-PAGE gels (Bio-Rad). Proteins from the SDS-PAGE gels were then transferred onto polyvinylidene fluoride membranes (Bio-Rad). Western blotting was performed with a mouse monoclonal anti-Talin antibody (clone 8d4, Sigma-Aldrich, T3287) at a dilution of 1:100 or a mouse monoclonal anti-ß-actin antibody (clone AC-74, Sigma-Aldrich, A2228) at a dilution of 1:1000 as the primary antibody and HRP-conjugated anti-mouse IgG antibody (eBioscience) at a dilution of 1:1000 as the secondary antibody.

## Measuring the expression level of xTalin1-EGFP and actin flow speed in xTalin1/2 KD cells

The expression of xTalin1-EGFP (molecules/μm²) was calculated by dividing the fluorescent intensity of EGFP in the measurement area by the average intensity of EGFP single-molecules that stuck on the grass surface outside of cell area with the same acquisition condition for

each experiment. The fluorescent intensity of EGFP was measured using MetaMorph software. The actin flow speed is the average of the flow speeds of ≥ 10 DyLight550-labeled actin or CF680R-labeled actin SiMS for each cell. The flow speed of actin SiMS was measured in lamellipodia with ≥4 frames (≥8 s).

## Simulation methods

**Chain model.** We model the Talin chain as a sequence of $N + 1$ beads, where each segment between successive beads represents one of $N$ Talin rod domains. Binding to integrin is simulated by a spring linking the first Talin bead to a bead fixed in space, while binding to F-actin is represented by a spring linking the last Talin bead to a bead that moves with the speed of retrograde flow, $v_{retro}$, in the horizontal direction as shown in Fig. 4a.

Each segment (subdomain) of the Talin chain can be in either a folded or an unfolded state. Folded rod domains are modeled as Hookean springs with a force magnitude of

$$F_{folded} = k_{folded}\left(l_{folded} - l_{folded,0}\right), \tag{5}$$

when the segment end beads are stretched at distance $l_{folded}$ beyond the equilibrium length $l_{folded,0} = 2\,nm$, and $k_{folded} = 10^4\,pN/\mu m$ is a spring constant typical of a Talin globular rod subdomain. The springs between Talin-actin and Talin-integrin are modeled with the same parameters as the folded Talin rod subdomains. For unfolded rod subdomains, the force magnitude when the beads are stretched by distance $l_{unfolded}$ is that of a freely jointed chain with an average end-to-end distance $l_{unfolded}$:

$$F_{unfolded} = (k_B T/b)L^{-1}\left(l_{unfolded}/b\,n_{talin}\right), \tag{6}$$

where $b = 0.38\,nm$ is an estimate of the Kuhn length (one amino acid)[47], $n_{talin} = 145$ is the typical number of amino acids per talin segment[9], $T = 300K$, and $L^{-1}$ is the inverse of the Langevin function[48,49].

Initially all rod subdomains are assumed to be folded and springs start at their equilibrium lengths aligned along a straight line with an orientation with respect to the horizontal specified by angle $\theta$ as shown in Fig. 4a, and assumed to lie on the plane of the figure. Thus, the total length of the chain including the Talin-integrin and Talin-actin bonds, $L_{tot}$, can be calculated over time as

$$L_{tot}(t) = \sqrt{L_{tot,0}^2 + 2L_{tot,0}v_{retro}t\cos\theta + v_{retro}^2 t^2}, \tag{7}$$

where $L_{tot,0} = (N + 2)l_{f,0}$ is the initial length of the chain.

**Force balance.** In the master equation method described below, tension along Talin is assumed to be uniform at every timestep. Thus, force balance determines the length of each segment and therefore the unfolding and unbinding rates. If there are $i$ unfolded rod subdomains, the lengths of the folded subdomains, $l_{folded}$, and the length of unfolded subdomains, $l_{unfolded}$, are determined by conservation of the total chain length,

$$L_{tot} = il_{unfolded} + (N + 2 - i)l_{folded}, \tag{8}$$

and uniform tension: $F_{folded} = F_{unfolded}$. These two equations are solved numerically together with Eqs. (4)−(7) at each timestep to determine the lengths $l_{folded}$ and $l_{unfolded}$.

**Unfolding and unbinding kinetics.** If there are $i$ unfolded rod sub-domains and tension is $F_i(t)$, each remaining folded Talin rod

subdomain unfolds with a rate dependent on Bell's law:[9,50]

$$k_i^{unfold}(t) = k_{unfold,0} \exp\left(F_i(t)\,\Delta x_{unfold}/k_B T\right), \tag{9}$$

where $k_{unfold,0}$ and $\Delta x_{unfold}$ are parameters specified for that domain. Correspondingly, the Talin chain unbinds from the integrin linker with a rate dependent on Bell's law:

$$k_i^{unbind}(t) = k_{unbind,0} \exp\left(F_i(t)\,\Delta x_{unbind}/k_B T\right), \tag{10}$$

with parameters $k_{unbind,0}$ and $\Delta x_{unbind}$.

**Master equation.** The master equation method is used to determine the probability $P_j$ of the Talin chain remaining linked to integrin with $j$ unfolded rod domains at time $t$:

$$\frac{dP_j(t)}{dt} = (M - j + 1)k_{j-1}^{unfold}(t)P_{j-1}(t) - (M - j)k_j^{unfold}(t)P_j(t) - k_j^{unbind}(t)P_j(t), \tag{11}$$

where $M$ is the initial number of subdomains, which can unfold, with $P_0(0) = 1$ and $P_j(0) = 0$ for $j > 0$. The probability $P_{unbound}(t)$ that the Talin has become unbound by time $t$ follows:

$$\frac{dP_{unbound}(t)}{dt} = \sum_{j=0}^{M} k_j^{unbind}(t)P_j(t). \tag{12}$$

We solved Eqs. (11) and (12) numerically, iteratively using a discrete time interval $\Delta t$, and the unfolding and unbinding rates from Eqs. (9) and (10) with the lengths and forces on the folded and unfolded domains as described in the *Force Balance* section. Specifically, allowing for multiple unfolding events within a single time interval between $t - \Delta t$ and $t$, the probabilities were approximated as

$$P_j(t) = \sum_{i=0}^{M} A_{ij}(t,\Delta t)P_i(t - \Delta t) - p_j^{unbind}(t,\Delta t)P_j(t - \Delta t),$$
$$P_{unbound}(t) = 1 - \sum_{j=0}^{M} P_j(t), \tag{13}$$

where $A_{ij}(t,\Delta t)$ is the probability of $j$-$i$ domains unfolding between time $t - \Delta t$ and $t$ given that there were $i$ domains unfolded at time $t - \Delta t$:

$$A_{ij}(t,\Delta t) = \binom{M - i}{j - i}\left(1 - p_i^{unfold}(t,\Delta t)\right)^{M-j} p_i^{unfold}(t,\Delta t)^{j-i}. \tag{14}$$

Here, $p_i^{unfold}(t,\Delta t)$ is the probability of a single domain unfolding between time $t - \Delta t$ and $t$ given that there are $i$ domains unfolded at time $t$:

$$p_i^{unfold}(t,\Delta t) = 1 - \exp\left(-k_i^{unfold}(t)\Delta t\right). \tag{15}$$

Similarly

$$p_i^{unbind}(t,\Delta t) = 1 - \exp\left(-k_i^{unbind}(t)\Delta t\right). \tag{16}$$

**Distributions of unbinding time, work done, and force on integrin.** From the master equation method, we determined the distributions of the unbinding times by numerically solving for $P_{unbound}(t)$. The derivative of $P_{unbound}(t)$ gives the probability distribution of unbinding times and distances $x = v_{retro}t$ corresponding to that time. We use the calculated distribution of unbinding distances to calibrate the value of parameter $\Delta x_{unbind}$ to match experimental measurements (Fig. S6A, B). The distribution of Talin tension at unbinding (i.e. the distribution of the maximum force) $P(F_{max})$ was calculated by binning the fraction of Talin chains that unbind from integrin with $j$ unfolded subdomains

between time $t - \Delta t$ and $t$, namely $p_j^{unbind}(t,\Delta t)P_j(t - \Delta t)$, into the corresponding force, $F_j(t)$. The average force of unbinding at time $t$ (i.e. average over the Talin ensemble) is written as

$$\overline{F(t)} = \frac{\sum_{j=0}^{M} F_j(t)P_j(t)k_j^{unbind}(t)}{\sum_{j=0}^{M} P_j(t)k_j^{unbind}(t)} \tag{17}$$

The average work done (by the ensemble) over time $t$ is equal to the integral of the component of the tension along the direction of retrograde flow times displacement. It was evaluated numerically at time $t = n\Delta t$ by:

$$\overline{W(t)} = \sum_{n=0}^{\frac{t}{\Delta t}} \sum_{j=0}^{M} F_j^x(n\Delta t)v_{retro}\Delta t P_j(n\Delta t). \tag{18}$$

**Model with different unfolding properties of Talin rod subdomains.** For a Talin chain where each rod subdomain $j$ has unique $k_{unfold,0}^j$ and $\Delta x_{unfold}^j$, the form of the master equation method is modified. In this case, each unique state, σ, of the chain needs to be accounted for rather than just the number of unfolded subdomains. Here σ is a binary array of size $M$, defining whether a specific rod subdomain is unfolded or not. The master equation can be written as

$$\frac{dP_\sigma(t)}{dt} = \sum_{j\,\text{in}\,\{\sigma'\}} k_{\sigma'}^{j,unfold}(t)P_{\sigma'}(t) - \sum_{j\,\text{in}\,\sigma} k_\sigma^{j,unfold}(t)P_\sigma(t) - k_\sigma^{unbind}(t)P_\sigma(t),$$
$$P_{unbound}(t) = 1 - \sum_\sigma P_\sigma(t), \tag{19}$$

where the first sum is over the $j$ subdomains which can unfold in $\{\sigma'\}$, which is the set of all states that take $\sigma' \to \sigma$ with a single unfolding event. The rate constant $k_{\sigma'}^{j,unfold}(t) = k_{unfold,0}^j \exp\left(F_{\sigma'}(t)\,\Delta x_{unfold}^j/k_B T\right)$, is the rate of unfolding of subdomain $j$ of a chain in state $\sigma'$ at time $t$, when the tension is $F_{\sigma'}(t)$. The second sum is over the $j$ subdomains which can unfold in σ, and $k_\sigma^{unbind}(t) = k_{unbind,0} \exp\left(F_\sigma(t)\,\Delta x_{unbind}/k_B T\right)$. The master Eqs. (19) can be solved in discrete time similar to Eqs. (13)–(15) and the distributions of unbinding time, work done, and force on integrin were calculated analogously to Eqs. (17) and (18).

**Individual Talin pulling simulations.** In addition to the master equation method we also run discrete Talin pulling simulations where seek to investigate the same model as described in the *Chain Model* section. The simulations solved Newton's laws for the beads representing the talin chain in the overdamped limit, with thermal fluctuations implicit in Bell's law and the entropic elasticity of unfolded domains. The evolution of the beads in these simulations follow the following equation:

$$\zeta \frac{d\mathbf{r}_i}{dt} = \mathbf{F}_{i,i-1} + \mathbf{F}_{i,i+1}, \tag{20}$$

where $\zeta$ is a bead friction coefficient and the force between beads $i$ and $i-1$, $\mathbf{F}_{i,i-1}$, depends on whether the segment between them represents a folded or unfolded rod subdomain:

$$\mathbf{F}_{i,i-1} = \begin{cases} -k_f\left(d_{i,i-1} - l_{f,0}\right)\hat{\mathbf{d}}_{i,i-1}, & \text{folded}, \\ (-k_B T/b)L^{-1}\left(d_{i,i-1}/bn_{talin}\right)\hat{\mathbf{d}}_{i,i-1}, & \text{unfolded} \end{cases} \tag{21}$$

Here $d_{i,i-1} = |\mathbf{r}_i - \mathbf{r}_{i-1}|$ and $\hat{\mathbf{d}}_{i,i-1}$ is the unit vector from bead $i-1$ to bead $i$. The integrin bead is fixed in space by setting its velocity to zero at each timestep. The actin bead moves with a constant speed $v_{retro}$ in the positive $x$-direction. The tension equilibrated along the chain quickly relative to the deformation by retrograde flow. We used the

Monte Carlo method to model the unfolding and unbinding events with rate constants given by Eqs. (9) and (10), replacing the force in these equations by the corresponding local tension. Once the Talin chain unbinds from integrin the simulation is stopped and the lifetime of the Talin chain is recorded. In order to build up accurate statistics many instances (~10,000) of these simulations need to be run and then averaged. These simulations use a timestep of $10^{-6}$ s.

**Comparison between the master equation and individual pulling simulations.** The main advantage of the discrete Talin pulling simulations is its simplicity that allows direct visualization of individual unfolding and unbinding events (Supplementary Movie 3). The main advantage of the master equation method is that the results represent the average behavior in significantly less simulation time. As a consistency check of the two simulation methods, we compared results from the master equation method to results from the individual Talin pulling simulations and found good agreement (Fig. S7B–D).

**Investigating impact of model assumptions.** To investigate how the assumptions of our model affect our main modeling result that mechanically unfolding Talin subdomains improve force transmission, we performed several additional simulations with the master equation, where we varied parameters of the model that had otherwise remained fixed up to this point.

We tested the implicit assumption in Fig. 4 that compliance of the integrin and actin layers was infinite. We relaxed this condition by adding two new springs to two newly added beads. The first new spring is between the integrin bead and a newly added bead represent the rest of integrin layer and substrate. This new substrate bead is now the only stationary bead in the simulation. The second new spring is between the actin bead and a newly added bead meant to represent the bulk actin network. This new actin network bead is the bead that moves at the constant retrograde flow speed. We find that as we decrease the spring constant of these newly added bonds, $k_3$, the average work done curve first flattens and then universally decreases (Fig. S8A). The transition for where the work done depends on the number of subdomains occurs for $k_3 > 10^2$–$10^3$ pN/μm. At values of $k_3 < 10^3$ pN/μm, however, we find that the majority of subdomains in Talin do not unfold by the time that it is 95% probable that the Talin chain has unbound (Fig S8C). Based upon this we argue that the value of $k_3$ should be at least of a similar magnitude to $10^3$ pN/μm, otherwise the ability of Talin to unfold is not doing much as the substrate and actin network are so soft that the Talin chain is unbinding before domains can unfold.

Finally, we tested the assumption of a slip-bond between integrin and Talin (an assumption motivated by prior Talin modeling studies[9,50]). Alternatively, the integrin-Talin linkage may have a catch-like component to it where the unbinding rate first decreases with force before ultimately failing at higher forces. To investigate the effect of a catch-bond for this linkage, we changed the unbinding properties of the integrin-talin bond to a catch-slip bond, using three different force-unbinding rate curves in place of Eqs. (10) (Fig. S8, left panel). The functional form used for the catch-bond curves assumes rapid equilibrium of the integrin-Talin bond between a fast and a slow-dissociating slip-bond state, with force increasing the probability of the slow state[51]. The transition rates between these two states and to the unbound state depend on force like $k_{ij}(F) = k_{ij}^0 e^{|F|x_{ij}/k_B T}$ where $k_{ij}^0$ is the force independent rate for transitioning from state $i$ to state $j$ and $x_{ij}$ represents the distance to the energy barrier for the $i$ to $j$ transition. State 0 corresponds to the unbound state, state 1 to the fast-dissociating bound state, and state 2 to the slow-dissociating bound state. The overall unbinding rate can then be written as $k_{unbind}(F) = \frac{k_{21}k_{10} + k_{12}k_{20}}{k_{12} + k_{21}}$, where all these rates depend on force. The unbinding rate at zero force was fixed at

the experimentally-measured rate and we varied the rate of the slow dissociating state to cover a range of depths of the minimum overall unbinding rate. Specifically, we set $k_{10}^0 = 0.194 \text{s}^{-1}$, $k_{12}^0/k_{21}^0 = 0.142$, $x_{10} = 0.0$, $x_{20} = 2.5 \text{nm}$, $x_{12} = 0.4 \text{nm}$, and $x_{21} = -1.6 \text{nm}$ with $k_{20}^0$ varied between $5.5 \times 10^{-6}$, $10^{-3}$, and $3 \times 10^{-3} \text{s}^{-1}$ (strongest to weakest catchbond). The strongest catch-bond unbinding rate curve in Fig. S8B (left) reaches an unbinding rate as low as that reported for Talin-actin under force[52]. We find that for all unbinding curves investigated, the average work still increases as a function of the number of subdomains that can unfold. Moreover, we find that the relative increase in the work done for the catch-slip bonds is higher than for the slip bond; indicating that this catch behavior strengthens our overall result. This is true even for the catch-slip bond with the shallowest unbinding rate minimum, which fits to the average displacement before unbinding given in experiment (dashed black line, Fig. S8B right panel). The catch-slip bonds with deeper minimum unbinding rates remain bound for much higher displacements than observed in experiment; so they likely do not reflect the real behavior of Talin.

### Statistics and reproducibility
All images are representative of at least five independent experiments. The number of samples ($n$) and statistic analysis methods are stated in the figure legends. Statistical analyses were performed in Excel and GraphPad Prism.

### Reporting summary
Further information on research design is available in the Nature Portfolio Reporting Summary linked to this article.

## Data availability
All the data supporting this study are available within the article and its Supplementary Information file. Source data are provided with this paper.

## Code availability
The code for simulations is available at https://github.com/davidmrutkowski/TalinStretching.

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

## Acknowledgements

We thank Ayako Kodera and Shu Yamamura for help with data analysis, and Aaron Hall, Danielle Holz and Shu Yamamura for discussions on mathematical simulations. This work was supported by Japan Society for the Promotion of Science KAKENHI Grant Number JP21K06150, JP22H04843, and JP21H05780 (S.Y.), JP22H00456 (N.W.), National Institutes of Health Grant R35GM136372, and by the National Science Foundation Lehigh Physics REU grant PHY-1852010 that supported K.L.

## Author contributions

S.Y. and Y.L. performed experiments and analyzed data. S.Y. and N.W. designed experiments. D.R., K.L. and D.V. performed simulations. S.Y., D.R., D.V. and N.W. wrote the manuscript. All other authors edited the manuscript.

## Competing interests

The authors declare no competing interests.
