## [Peer review file · Nature Communications]

REVIEWER COMMENTS

Reviewer #1 (Remarks to the Author):

The authors provide an excellent revision in which they address all reviewer comments to my full satisfaction. This is a great study and I look forward to seeing it in press.

Reviewer #2 (Remarks to the Author):

For the sake of clarity I will organize my comments with reference to the points raised in the initial review:

- 1) The authors state that " k_c was calculated from the average clutch duration (Fig. 2D)." However, 2D shows the distribution of jump lengths for N-terminally tagged α Talin1. Unfortunately, it is still not clear how k_c was calculated. Since I still do not understand how k_c , and by extension k_b , are calculated, it is not possible to make further headway on this issue. However, as noted in the original review, if the authors are basing their kinetic modeling on the duration of a sub-state in truncated time series data, it is important to take into account the fact that long-lived sub-states are less likely to be observed simply because the time series is likely to be interrupted by a separate process (here, photobleaching).
- 2) The point I was trying to make in the original review is that the authors cannot easily conclude that α Talin1 molecules that are stationary also experience zero tension. In Driscoll, the authors use a variety of measurements to indicate that tension on F-actin persists even in circumstances in which the F-actin flow velocity is close to zero (see for example Fig. 1C). The authors of that paper conclude that under these circumstances most of the force generation results from myosin, which is resisted by linkages to talin, resulting in a force balance between the two. These data are pertinent in that if the actin is stationary, then the talin c-terminus should not be moving either, even though force is present. The other two references reinforce this point.
- 3) Please see doi.org/10.1101/2023.01.31.526409. However, the general point that spectrin unfolding may substitute for talin unfolding is reasonable.
- 4) It is puzzling to the reviewer that the authors argue that the Yamashiro et al. paper illustrates that F-actin flows with uniform velocity. Figure 6C in that study, for example, shows that this is clearly not the case near cell adhesions. The simulation in Fig. 6A is not helpful in that so far as I can determine in this simulation the rate of uniform actin flow is switched rapidly between 0 and 40 nm/s, and at a very high frequency relative to the other rates in this system. It is not surprising that the authors observe time-averaged behavior in this circumstance. Instead, the question of interest, which I do believe the authors should seriously consider, is what the effect on their study would be if they consider a circumstance in which individual F-actin filaments move different velocities. The other two simulations are helpful in addressing the corresponding aspects of the previous critique.
- 5) Unless I somehow missed it, the authors still do not cite the relevant study from del Río Hernández. Also, regardless of the authors' evident distaste for the study, the paper by Margadent et al. is in the literature and does report that talin extends in cells. To my understanding, failure to cite prior relevant work is inconsistent with the editorial standards of Nature Communications.

6) The argument that "Domain unfolding sustains the force of individual molecular connections between integrin and actin for longer times (this is the meaning of better force transmission)..." is a direct correlate of the points made in the study by Jie Yan's group.

Reply to reviewers' comments (NCOMMS-23-19213-T, Yamashiro et al.):

Response to Reviewer #1:

The authors provide an excellent revision in which they address all reviewer comments to my full satisfaction. This is a great study and I look forward to seeing it in press.

Thanks to your comments in the initial review, we have been able to improve our paper very much. We appreciate your support.

Response to Reviewer #2:

For the sake of clarity I will organize my comments with reference to the points raised in the initial review:

1) The authors state that “ k_c was calculated from the average clutch duration (Fig. 2D).” However, 2D shows the distribution of jump lengths for N-terminally tagged xTalin1. Unfortunately, it is still not clear how k_c was calculated. Since I still do not understand how k_c , and by extension k_b , are calculated, it is not possible to make further headway on this issue. However, as noted in the original review, if the authors are basing their kinetic modeling on the duration of a sub-state in truncated time series data, it is important to take into account the fact that long-lived sub-states are less likely to be observed simply because the time series is likely to be interrupted by a separate process (here, photobleaching).

The rate constant k_c was calculated from the reciprocal of the average clutch duration of EGFP-xTalin1 SiMS switching motion from stationary to flowing, assuming that switching from the clutch state to the flowing state was a first-order reaction. The average clutch duration was calculated by dividing the distance between the position at the stationary state and the first position where the flow is detected (Δx , shown in Fig. 2D) by the flow speed for each EGFP-xTalin1 SiMS. The rate constant k_b was determined from k_c and the ratio of frequency at which stationary xTalin1-EGFP switched to a flowing state to the frequency at which back-and-forth motion of xTalin1-EGFP was observed. We have added an explanation of the calculation of k_c and k_b in the Materials and Methods section (p34, the 2nd paragraph).

As we mentioned in our previous reply to the reviewer's comment #1, we counted the number of events where xTalin1 SiMS switched its behavior from stationary to flowing to obtain the frequency per unit time to measure the rate constant k_a . The rate

constant k_c and k_b were calculated as described in the previous paragraph. In these measurements of the rate constants k_a and k_b , the frequency of events occurring per unit time is not affected by photobleaching. In these measurements of the rate constants k_c , the average clutch duration calculated from the distance (Δx) and the flow speed is also not affected by photobleaching.

2) The point I was trying to make in the original review is that the authors cannot easily conclude that α Talin1 molecules that are stationary also experience zero tension. In Driscoll, the authors use a variety of measurements to indicate that tension on F-actin persists even in circumstances in which the F-actin flow velocity is close to zero (see for example Fig. 1C). The authors of that paper conclude that under these circumstances most of the force generation results from myosin, which is resisted by linkages to talin, resulting in a force balance between the two. These data are pertinent in that if the actin is stationary, then the talin c-terminus should not be moving either, even though force is present. The other two references reinforce this point.

In Driscoll *et al.* (*PNAS*, 117, 32413-32422, 2020), regions with F-actin flow velocities close to zero are clustered in the inner regions away from the leading edge (see Fig. 1B, which shows the heat maps for actin speed). Furthermore, the caption of Fig. 1C (Driscoll *et al.*) describes binned pixelwise correlations of talin tension-sensor FRET vs. actin speed. Thus, the data demonstrated in Driscoll *et al.* indicate neither the velocity of individual actin filaments nor the existence of arrested actin filaments in flowing F-actin networks.

Furthermore, to address the reviewers' comments 2 and 4, we have added new data of single-molecule analysis of F-actin motion within mature focal adhesions (FAs) (new Suppl. Fig. S1). We performed SiMS imaging of DyLight550-labeled actin (DL-actin) at 500 ms intervals for 10 sec in XTC cells. We then analyzed actin SiMS within mature FAs by using our nanometer-scale displacement measurements (Yamashiro *et al.*, *MBoC* 25, 2014). Our new data showed that all observed actin SiMS moved at a constant speed over mature FAs (new Suppl. Fig. S1, B and C). The standard deviation of the difference between the measured displacement of actin SiMS and the linear approximation of the speed was ± 10.9 nm, which is comparable to the localization error of our previously reported nanometer-scale displacement measurement (*MBoC*, 2014). Also, we previously demonstrated that F-actin flowing over thin elongated FAs

move at similar speeds to those flowing near but outside FAs in lamellipodia (Yamashiro *et al.*, *MBoC*, 2014, Fig. 7). These results indicate that the actin network moves at a constant speed over FAs.

In addition to these single-molecule analyses of F-actin in FAs, we have repeatedly shown that almost all F-actin moved with the retrograde flow in lamellipodia containing FAs (Figs 6 and 7, Suppl Fig. S8 and Movies 8-10 in Yamashiro *et al.*, *MBoC*, 2014; Fig. 1F and Video 2 in this study). Actin speckles that hardly move have been rarely observed in our series of actin SiMS observations. In contrast, this study reveals that 50.0% of xTalin SiMS exhibited a stationary motion in lamellipodia where the flowing actin network exists (Fig. 1A, C, D; Suppl. Table 1). Therefore, it is unlikely that stationary Talin molecules and arrested actin filaments exist in a force-balanced state. We have added this data in the new Suppl. Fig. S1 and explained the results (p6, the 2nd paragraph). We have replaced previous Suppl. Fig. S1 to S6 with new Suppl. Fig. S2 to S7.

3) Please see doi.org/10.1101/2023.01.31.526409. However, the general point that spectrin unfolding may substitute for talin unfolding is reasonable.

We appreciate your understanding of our data. We also thank you for informing us of the interesting preprint paper.

4) It is puzzling to the reviewer that the authors argue that the Yamashiro et al. paper illustrates that F-actin flows with uniform velocity. Figure 6C in that study, for example, shows that this is clearly not the case near cell adhesions. The simulation in Fig. 6A is not helpful in that so far as I can determine in this simulation the rate of uniform actin flow is switched rapidly between 0 and 40 nm/s, and at a very high frequency relative to the other rates in this system. It is not surprising that the authors observe time-averaged behavior in this circumstance. Instead, the question of interest, which I do believe the authors should seriously consider, is what the effect on their study would be if they consider a circumstance in which individual F-actin filaments move different velocities. The other two simulations are helpful in addressing the corresponding aspects of the previous critique.

In our previous study, we observed that F-actin flow velocities vary locally

near mature FAs, whereas actin speckle speeds are similar in the center of mature FAs (Yamashiro *et al.*, *MBoC* 2014, Fig. 7A). To confirm our previous results with higher resolution, we have added new data of single-molecule analysis of F-actin motion within mature FAs as in our reply to Reviewer#2's comment 2. In the new Suppl. Fig. S1C, we show displacement measurements of actin SiMS within FAs in a series of DL-actin SiMS images with a low localization error of 10.9 nm with 500-ms temporal resolution. The data indicates that all observed actin SiMS continuously moved at similar flow speeds (new Suppl. Fig. S1C). As mentioned in our reply to Reviewer#2's comment 2, actin SiMS also flow at a constant speed over thin elongated FAs. Based on our results from measuring individual filaments, we conclude that actin filaments form a connected network via cross-linkers and move as a single unit with the retrograde actin flow over FAs.

As we discussed in our previous study (Yamashiro *et al.*, *MBoC* 2014, Fig. 7A), remodeling of the local actin network occurs near mature FAs. We have revealed part of the mechanism by which remodeling of the actin network occurs near FAs, and are currently preparing a manuscript for submission. As the reviewer pointed out, the flow velocity is not uniform in the limited regions near mature FAs. Nevertheless, in most of the actin network, actin filaments continuously flow at similar speeds. We have added an explanation of F-actin flow velocities near and over mature FAs in the text (p6, the 2nd paragraph).

We have removed previous Suppl. Fig. S6A, and replaced previous Suppl. Fig. S6B and S6C with new Suppl. Fig. S7A and S7B, respectively. We have revised the legend of the new Suppl. Fig. S7 and the text to reflect these changes.

5) Unless I somehow missed it, the authors still do not cite the relevant study from del Río Hernández. Also, regardless of the authors' evident distaste for the study, the paper by Margadent et al. is in the literature and does report that talin extends in cells. To my understanding, failure to cite prior relevant work is inconsistent with the editorial standards of Nature Communications.

We have cited the study from the del Río Hernandez lab as the reference #25 (p8, the 3rd paragraph and p15, the 1st paragraph). We had already cited the article by Margadent et al in Discussion (p16, the 2nd paragraph) as the reference #38.

6) *The argument that “Domain unfolding sustains the force of individual molecular connections between integrin and actin for longer times (this is the meaning of better force transmission)...” is a direct correlate of the points made in the study by Jie Yan's group.*

Jie Yan's group performed kinetics simulations and reported that unfolding of Talin rod subdomains can prevent force accumulation during Talin elongation and maintain the force transmitted via Talin below 10 pN (Yao *et al.*, *Nat Commun*, 7:11966, 2016). Based on these results, Yao *et al.* propose that Talin serves as a force buffer. However, the concept of a force buffer in this work appears to indicate maintenance of a static mechanical equilibrium, without considering the unavoidable extension and dissociation of molecular connections or the work done per connection. In contrast, as we mentioned in the previous reply to the reviewers' comments, we suggest that the unfolding of Talin rod subdomains promotes force transmission associated with the clutch by increasing the average clutch duration (Fig. 4). We have added a description about the article by Yao *et al* (the reference #9) in more detail in Discussion (p16, the 1st paragraph).

REVIEWER COMMENTS

Reviewer #2 (Remarks to the Author):

1) I now understand the method by which the authors calculate their kinetic rate constants, which is good, and am less concerned about the effects of photobleaching. However, I still do not fully understand the statement: "The rate constant k_b was determined from k_c and the ratio of frequency at which stationary xTalin1-EGFP switched to a flowing state to the frequency at which back-and-forth motion of xTalin1-EGFP observed." Do the authors mean the frequency of backward jumps?

2) Other aspects of the authors' kinetic model still give rise to concerns. There is no reason to assume, as the authors do, that switching from the clutch state to the flowing state is a first-order reaction. This limitation should be made clear.

3) For the calculation of k_c , the authors appear to use an uncorrected jump distance for the N-terminal EGFP tag on talin of ~ 50 nm (Fig. 2D). However, they also report apparent jumps of ~ 35 nm for a C-terminal tag on talin, which in the authors model should not undergo jumps. This suggests that the jump distances the authors measure are inflated by measurement error, which is present should be corrected for. This matters, for two reasons. First, it suggests that the "jumps" may be ~ 15 nm, which is shorter than the length of talin, calling into question the correctness of their assignment. Second, it would alter the inferred kinetics by a factor of 2 to 3.

4) Regarding the comparison with Driscoll et al., the authors would be prudent to note that the actin cytoskeletons of different cell types may behave differently, and in particular that the coordinated retrograde flows that are described in this study may not characterize all regions of all cell types.

4) Regarding the new Figure S1, the authors report tracking data for a total of 18 actin molecules, which does not rise above the level of anecdote.

5) The largest problem with the manuscript is that the authors continue to argue that the authors' preferred model corresponds to a novel finding: Quoting the abstract: "This study reveals a new mode of force transmission, in which stochastic molecular stretching bridges two cellular structures moving at different speeds." This is not a new idea, and follows directly from studies by Margadent et al., Haining et al., and Yao et al. (and possibly others I may have overlooked). Although these studies are cited, the relevance of these papers to the present manuscript is not made clear to the reader.

Reply to reviewers' comments (NCOMMS-23-19213-A, Yamashiro et al.):

Response to Reviewer #2:

1) I now understand the method by which the authors calculate their kinetic rate constants, which is good, and am less concerned about the effects of photobleaching. However, I still do not fully understand the statement: "The rate constant k_b was determined from k_c and the ratio of frequency at which stationary xTalin1-EGFP switched to a flowing state to the frequency at which back-and-forth motion of xTalin1-EGFP observed." Do the authors mean the frequency of backward jumps?

Yes, we measured the frequency of backward motion of xTalin1-EGFP as a back-and-forth motion. The ratio of the frequencies with which the clutching xTalin1 switches to a flowing motion (marked with an asterisk) and to a back-and-forth motion (marked with a dagger) was determined from the data in TableS2 (p47). As the frequency with which the clutching xTalin1 switches to a flowing motion is about 1.82 times of the frequency of switching to a back-and-forth motion, we approximated k_b by dividing $k_c = 0.680 \text{ s}^{-1}$ by the ratio of these frequencies. We have added an explanatory sentence to the legend and symbols to mark the data in TableS2 (p47).

As we had described in the text (the last paragraph of p7 to the 2nd paragraph of p8), our nanometer-scale displacement analysis of EGFP-tagged xTalin1 revealed two types of movement: jumping and back-and-forth motions. Single-molecules of xTalin1-EGFP that behave with a back-and-forth motion move in the actin flow direction from a stationary state, but return to their original position (Fig. 2, E and F). To avoid confusion, we have unified to indicate "back-and-forth motion" in Table S2, Fig. 2E and the legend of Fig. 2 (it had been written as "Swinging speckle in a flow direction" in the previous manuscript). In addition, we have found our mistake in the previous Fig. 2F, which has been corrected.

2) Other aspects of the authors' kinetic model still give rise to concerns. There is no reason to assume, as the authors do, that switching from the clutch state to the flowing state is a first-order reaction. This limitation should be made clear.

We agree with the reviewer that the switching of Talin from the clutch state to the flowing state may not be a first-order reaction. We have revised the Materials and Methods section (p35, the 2nd paragraph) to describe that we approximated k_c from the reciprocal of the average clutch duration of EGFP-xTalin1 SiMS, as the average frequency at which the clutching xTalin1 transition to a flowing state occurs per second.

3) For the calculation of k_c , the authors appear to use an uncorrected jump distance for the N-terminal EGFP tag on talin of ~ 50 nm (Fig. 2D). However, they also report apparent jumps of ~ 35 nm for a C-terminal tag on talin, which in the authors model should not undergo jumps. This suggests that the jump distances the authors measure are inflated by measurement error, which is present should be corrected for. This matters, for two reasons. First, it suggests that the “jumps” may be ~ 15 nm, which is shorter than the length of talin, calling into question the correctness of their assignment. Second, it would alter the inferred kinetics by a factor of 2 to 3.

We thank the reviewer for pointing out this critical issue. We estimated the accuracy for our measurement of the xTalin1 jumping distance. The criteria are described in the Materials and Methods section (p33-p34) as follows: “the distance moved in the flow direction from the origin (the stationary position) for each frame was plotted against time. The time when the distance exceeds the localization error (18.6 nm) three times continuously in the flow direction was defined as a switching to a flowing motion.” The graph below shows the expected probability (y-axis) that the jumping distance can be measured for xTalin1 SiMS that jump at different distances (x-axis).

As shown in the above graph, when xTalin1 jumps 40 nm or more, the jumping distance can be measured with more than 75% sensitivity. Therefore, we performed a valid measurement for the N-terminal EGFP tag on xTalin1 with $\Delta x > 40$ nm, which account for the majority of the N-tagged xTalin1 SiMS measured, shown in Fig. 2C. We have added the sensitivity of the jumping distance measurement in the new Suppl Fig. S3 (p53) and the explanation of the data in the text (p8, the 1st paragraph), the legend of Fig. 2C (p25) and the Materials and Methods section (p34, the 1st paragraph). We have replaced previous Fig. S3 to S7 with new Suppl. Fig. S4 to S8.

On the other hand, the Δx for the C-terminal tag on xTalin1 (xTalin1-EGFP, Fig. 2C) was 31.1 nm on average, suggesting that the C-terminal EGFP tag on xTalin1 might shift slightly

in the flow direction. This is presumably due to the conformation change of Talin from a 15-nm globular to a ~60-nm open form upon F-actin binding (Dedden *et al.*, *Cell* 179, 120-31, 2019). Regarding the N-terminal tag Δx , we estimate from the cryo-EM study (Dedden *et al.*, *Cell*, 2019) that the conformation change of the N-terminal FERM domain upon dissociation from integrin is negligible. We have added the discussion for the C-terminal tag Δx in the text (p8, the 1st paragraph) and the legend of Suppl. Fig. S3 (p53).

4) Regarding the comparison with Driscoll et al., the authors would be prudent to note that the actin cytoskeletons of different cell types may behave differently, and in particular that the coordinated retrograde flows that are described in this study may not characterize all regions of all cell types.

In this study, we have revealed motions of Talin and actin molecules through observations with superior time and spatial resolution to the previous studies. In the study by Driscoll *et al.*, the actin cytoskeleton was visualized with a very high density of a fluorescent actin probe (see below, green shows actin. The image was taken from Movie S1 of Driscoll *et al.*, *PNAS*, 117, 2020) and analyzed using quantitative Fluorescent Speckle Microscopy (qFSM).

In particle tracking analysis, if the density of probes is high, accurate measurements cannot be made no matter what method is used (Chenouard *et al.*, *Nat Method* 11, 281-9, 2014; Vallotton *et al.*, *Traffic* 18, 840-52, 2017), as we discussed in the Discussion section (p17, the 2nd paragraph). In this study, we took particular care to accurately track individual actin and Talin SiMS. For example, our SiMS data show that the behaviors of Talin molecules are clearly divided into two modes, stationary and flowing, which has not been reported in previous quantitative imaging

Redacted

of Talin (Brown *et al.*, *J Cell Sci* 119, 5204-14, 2006; Hu *et al.*, *Science* 315, 111-5, 2007; Margadant *et al.*, *PLoS Biol* 9, e1001223, 2011; Stutchbury *et al.*, *J Cell Sci* 130, 1612-24, 2017). To discuss actin structures in other cell types and regions, it is necessary to clarify the movement of F-actin in all cell types using our SiMS method, but that is not the scope of our current study. Therefore, we decline to add such discussion.

It can often be difficult to accurately measure the retrograde flow velocity by qFSM, because automated object tracking may misidentify individual speckles when they are densely packed. The fidelity of the qFSM method had been argued between leading scientists (Vallotton & Small, *J Cell Sci*, 122, 1955-8, 2009; Danuser, *J Cell Sci*, 122, 1959-62, 2009). Dr. Vallotton is one of the developers of the qFSM technique (Vallotton *et al.*, *PNAS*, 2004), but he has repeatedly warned about errors in automated speckle tracking caused by high-density probes (Vallotton & Small, *J Cell Sci*, 2009; Vallotton *et al.*, *Traffic*, 2017). Compared to the qFSM, our SiMS method, which enables to monitor individual actin filaments directly, is clearly more precise for quantitative analysis. In our previous study, we measured the retrograde flow with the highest precision velocity measurement, in which actin SiMS displacements of 100-150 nm for 3-4 s are sufficient for reliable measurement (Yamashiro *et al.*, *MBoC*, 2014). By using the method, our previous SiMS study revealed locally different F-actin flows in XTC cells; i.e., F-actin flow velocities vary locally near mature FAs, whereas actin speckle speeds are similar in the center of mature FAs (Yamashiro *et al.*, *MBoC*, 2014). In this study, we confirmed that F-actin flowed at a constant speed in mature FAs (Suppl. Fig. S1). Such relation between F-actin flow and FAs was first uncovered with the high resolution and precision of our SiMS microscopy. It is not the scope of this study to characterize how this is the case in other cell types.

5) Regarding the new Figure S1, the authors report tracking data for a total of 18 actin molecules, which does not rise above the level of anecdote.

We reproducibly observed the data showing actin SiMS moved at a constant speed over mature FAs as shown in Suppl. Fig. S1. We have added the data of other four cells in the new Suppl. Fig. S1 (p49-p51), showing that all observed actin SiMS moved at a constant speed over mature FAs.

6) The largest problem with the manuscript is that the authors continue to argue that the authors' preferred model corresponds to a novel finding: Quoting the abstract: "This study reveals a new mode of force transmission, in which stochastic molecular stretching bridges two cellular structures moving at different speeds." This is not a new idea, and follows directly from studies by Margadent et al., Haining et al., and Yao et al. (and possibly others I may have overlooked). Although these studies are cited, the relevance of these papers to the present manuscript is not made clear to the reader.

We have removed "a new mode" from the last sentence in the abstract and revised it as; "This study reveals a force transmission mechanism, in which stochastic molecular stretching

bridges two cellular structures moving at different speeds.” (p2, the last sentence)

As we mentioned in our first reply to reviewers’ comments, the study by Margadant *et al* (*PLOS Biol*, 9, e1001223, 2011) does not show convincing evidence of single-molecule imaging and claims conclusions based on problematic data. It is obvious that Margadant *et al* observed aggregates of a high density of fluorescent molecules. We hesitate to strongly deny the Margadant *et al* paper in our manuscript, but we describe the difference between our study and previous studies including Margadant *et al*, as “*Our SiMS data show that the behaviors of Talin molecules are clearly divided into two modes, flowing and stationary, which has not been reported in the previous quantitative imaging of Talin^{14,38,39}. In the previous studies, the quantification may have failed to detect the two modes due to measuring the mixed population^{14,38,39}.*” (p.17, the 2nd paragraph, ref.39 is Margadant *et al*).

We have added one paragraph in the Discussion section, to describe the differences of this work from the previous *in vitro* single-molecule studies by Yao *et al* (ref. 9) and Haining *et al* (ref. 26) as, “*Talin has been extensively studied for its force-induced unfolding property by in vitro single-molecule experiments^{6,9,26}. The unfolding of Talin rod subdomains has been proposed to act as a mechanosensor to control force-dependent interactions with its binding proteins^{5,6,26}. The unfolding of Talin rod domain may also act as a force buffer that maintains mechanical equilibrium at low forces over a wide range of Talin extensions⁹. The Talin extension fluctuations in response to changes in force have been suggested to be responsible for Talin’s mechanosensing functions⁹. However, the lifetime and kinetics of the linkage by Talin in cells were unknown, and the role of the unfolding of Talin in force transmission has not been critically examined in the previous studies^{6,9,26}. Owing to the capability of the SiMS approach, our present study directly visualized stochastic, transient linkage of Talin in live cells. Furthermore, our simulations provide evidence that stochastic unfolding of subdomains facilitates the force transmission between the flowing F-actin network and the substrate by extending the clutch duration. Our study thus elucidates the direct link between molecular biophysics and cellular mechanics.*” (p14, the 2nd paragraph to p15)

REVIEWERS' COMMENTS

Reviewer #2 (Remarks to the Author):

1) The authors state that:

"We previously reported that the velocity of the retrograde flow varies locally near mature FAs, whereas the flow speeds of actin SiMS are similar in the center of mature FAs. We confirmed that all observed F-actin in FAs continuously moved at similar flow speeds measured with a low localization error of 11.7 nm with 500-ms temporal resolution (Fig. S1C). These results support that except for near mature FAs, the F-actin network constantly moves as a single unit over FAs."

I do not understand why there should be a difference for actin motion "near" focal adhesions (FAs) vs. "over" them. One possibility is that the authors wish to reconcile the results of their previous publication with their current, favored model. This concern could be addressed, at least in part, by explaining the criteria that the authors used to select puncta, and cellular subregions, for tracking in figure S1.

2) The authors are still seemingly reluctant to acknowledge prior results. Quoting the last line of the abstract: "This study reveals a force transmission mechanism, in which stochastic molecular stretching bridges two cellular structures moving at different speeds." The authors can claim that their data lend support to the consensus model for how talin transmits force between F-actin and adhesions. However, "reveals" is not accurate, as it implies that the mechanism proposed by the authors was unknown prior to the study, which is not the case. I once more urge the authors to transparently present prior studies relevant to the present work.

Reply to reviewers' comments (NCOMMS-23-19213-B, Yamashiro et al.):

Response to Reviewer #2:

1) "We previously reported that the velocity of the retrograde flow varies locally near mature FAs, whereas the flow speeds of actin SiMS are similar in the center of mature FAs. We confirmed that all observed F-actin in FAs continuously moved at similar flow speeds measured with a low localization error of 11.7 nm with 500-ms temporal resolution (Fig. S1C). These results support that except for near mature FAs, the F-actin network constantly moves as a single unit over FAs."

I do not understand why there should be a difference for actin motion "near" focal adhesions (FAs) vs. "over" them. One possibility is that the authors wish to reconcile the results of their previous publication with their current, favored model. This concern could be addressed, at least in part, but explaining the criteria that the authors used to select puncta, and cellular subregions, for tracking in figure S1.

The criteria for DyLight550-labeled actin (DL550-actin) SiMS that we tracked in Fig. S1B and C is DL550-actin SiMS in the overlapping region with the signal of EGFP-paxillin, a marker of focal adhesions. We have added explanation of the criteria for actin SiMS to be analyzed to the legend of Supplemental Fig. S1B.

We used an unattenuated 100-W mercury illumination to acquire DL550-actin SiMS as described in the legend of Fig. S1. To minimize photodamage due to the strong illumination, we restricted the illuminated area to the cellular region containing focal adhesions with the microscope field diaphragm.

We have revised the paragraph to describe more clearly as *"We verified the uniform actin flow in lamellipodia containing FAs by using the nanometer-scale displacement measurement of actin SiMS¹⁶ (Fig. S1). We previously reported that the actin SiMS approaching the FA area slow down within 0.5 μm of the outer edge of mature FAs, whereas the flow speeds of actin SiMS are uniform in the center of mature FAs¹⁶. We confirmed that all observed F-actin in FAs continuously moved at similar flow speeds measured with a low localization error of 11.7 nm with 500-ms temporal resolution (Fig. S1C). These results support that the F-actin network constantly moves as a single unit in the center of FAs."* (p.6, the 2nd paragraph).

2) *The authors are still seemingly reluctant to acknowledge prior results. Quoting the last line of the abstract: "This study reveals a force transmission mechanism, in which stochastic molecular stretching bridges two cellular structures moving at different speeds." The authors can claim that their data lend support to the consensus model for how talin transmits force between F-actin and adhesions. However, "reveals" is not accurate, as it implies that the mechanism proposed by the authors was unknown prior to the study, which is not the case. I once more urge the authors to transparently present prior studies relevant to the present work.*

We have revised the last sentence in the abstract as; *"This study elucidates a force transmission mechanism, in which stochastic molecular stretching bridges two cellular structures moving at different speeds."* (p2, the last sentence). We also removed "new" and "novel" about our study from the main text according to a suggestion from the editor.

We had already described the differences of this work from the previous single-molecule studies by Yao *et al* (ref. 9) and Haining *et al* (ref. 26) and the previous Talin imaging studies by Hu *et al* (ref. 14), Brown *et al* (ref. 38) and Margadant *et al* (ref. 39) in Discussion.